# Actomyosin and the MRTF-SRF pathway downregulate FGFR1 in mesenchymal stromal cells

Jip Zonderland[1], Silvia Rezzola[1] & Lorenzo Moroni [1✉]

Both biological and mechanical signals are known to influence cell proliferation. However, biological signals are mostly studied in two-dimensions (2D) and the interplay between these different pathways is largely unstudied. Here, we investigated the influence of the cell culture environment on the response to bFGF, a widely studied and important proliferation growth factor. We observed that human mesenchymal stromal cells (hMSCs), but not fibroblasts, lose the ability to respond to soluble or covalently bound bFGF when cultured on microfibrillar substrates. This behavior correlated with a downregulation of FGF receptor 1 (FGFR1) expression of hMSCs on microfibrillar substrates. Inhibition of actomyosin or the MRTF/SRF pathway decreased FGFR1 expression in hMSCs, fibroblasts and MG63 cells. To our knowledge, this is the first time FGFR1 expression is shown to be regulated through a mechanosensitive pathway in hMSCs. These results add to the sparse literature on FGFR1 regulation and potentially aid designing tissue engineering constructs that better control cell proliferation.

[1] Complex Tissue Regeneration Department, MERLN Institute for Technology-Inspired Regenerative Medicine, Maastricht University, Maastricht, the Netherlands. ✉email: l.moroni@maastrichtuniversity.nl

Understanding proliferation is critical for regenerative medicine approaches. It is important to understand and control proliferation both in vitro, to obtain a sufficient number of cells for proliferation, and in vivo, to control cell growth and regeneration. Both biological signals provided by growth factors as well as mechanical signals from the cellular microenvironment have been shown to influence proliferation of cells[1,2]. In this study, we investigated the cross-talk between these different signals.

Several growth factors are well known for their proliferation inducing abilities. Arguably, the most well-studied of these is basic fibroblast growth factor (bFGF). bFGF is known to increase proliferation rates in a wide variety of cell types and has anti-apoptotic effects, while maintaining or enhancing differentiation- and regeneration potential[3]. bFGF can bind to 7 FGF receptors (FGFR; coming from 4 FGFR genes, FGFR1-4). All FGFRs are tyrosine kinase receptors that can activate a variety of pathways, including the RAS-MAPK, PI3K-Akt, PLCγ, and STAT pathways[4]. FGFRs are also important for regenerative medicine purposes. FGFR1 and 2 have been shown to be involved in adipo- and osteogenic differentiation in hMSCs[5,6]. FGFR3 is highly expressed in chondrocytes and involved in chondrogenesis[7]. Only FGFR1 has been shown to be involved in hMSCs proliferation[8], while the other receptors remain unstudied in this regard. For this reason, here we focused on the regulation of FGFR1 expression.

Very little is known about the regulation of any FGFR. YAP knockdown has been shown to decrease FGFR1 expression in lung cancer cells[9] and neurospheres[10]. Also, integrin α6 has been shown to regulate FGFR1[10]. Both YAP and integrins play an important role in mechanosensing[11,12], hinting at a potential mechanosensitive regulation of FGFR1.

Cells adhere to their surrounding matrix or culture substrate through integrins[13]. When enough force is applied, integrin clusters can bind to the actin cytoskeleton through large protein complexes called focal adhesions[14]. On the other end, actin filaments can be attached to other focal adhesions, or to the nucleus[15]. Between these attachment points, force can be generated by actin-myosin filaments to generate cellular tension[16]. A large variety of cellular processes are regulated by cellular tension, including proliferation[11,17–19], differentiation[20–22], and migration[23]. Different transcription factors have been shown to orchestrate these changes in behaviors, of which serum response factor (SRF) is a well-studied example. When globular actin concentrations are low in the cytoplasm, myocardin related transcription factor (MRTF) A or B enters the nucleus and binds to SRF to start transcribing target genes[24].

As transplanted cells for regenerative medicine inevitably end up in a 3D environment, we wanted to investigate the potential mechanosensitive regulation of FGFR1 and response to bFGF in 3D. Previously, we have shown that hMSCs reduce cellular tension in 3D microfibrillar substrates and other 3D environments[25]. Thus, to investigate the effect of bFGF in 3D and to potentially find leads on FGFR1 regulation, we started by investigating the response of hMSCs to bFGF in a 3D microfibrillar environment. In solution, bFGF, like most growth factors, is highly unstable and loses activity after 24-48 h[3,26]. Covalently coupling bFGF to scaffolds has been shown to enhance stability while maintaining signaling activity[27–29]. Therefore, we tested the response to bFGF in 3D for both soluble bFGF and covalently bound bFGF on microfibrillar substrates.

Here, we have found that hMSCs do not respond to soluble or covalently bound bFGF when cultured on microfibrillar substrates, while fibroblasts do. hMSCs, but not fibroblasts, down-regulate FGFR1 expression when cultured on microfibrillar substrates. We show that FGFR1 expression is regulated through the mechanosensitive proteins actin-myosin and MRTF/SRF.

Further, the inhibition of the MRTF/SRF pathway made hMSCs irresponsive to bFGF on tissue culture plastic (TCP) and down-regulated FGFR1 in hMSCs and fibroblasts.

## Results

**Fibroblasts, but not hMSCs, respond to bFGF functionalized microfibrillar substrates.** To study the effect of bFGF on hMSCs in a 3D environment, we set out to covalently couple bFGF to microfibrillar scaffolds. Microfibrillar substrates with a thickness of 50 μm and $0.99 \pm 0.18$ μm average fiber diameter were produced by electrospinning 300PEOT45PBT55 (Supplementary Fig. 1). The ester bond in the polymer was opened using 0.5 M NaOH to expose carboxyl groups on the surface of the scaffold. 1-ethyl-(dimethylaminopropyl)-carbodiimide (EDC) - N-hydroxysuccinimide (NHS) chemistry was used to covalently couple the free amine groups of proteins to the surface of the scaffold. To validate our approach, we first coupled fluorescent FITC-labeled bovine serum albumin (BSA) to the microfibrillar substrates. A $27 \pm 1$ fold ($p < 0.001$) higher fluorescent signal was observed when BSA-FITC was added after EDC-NHS, than when BSA-FITC was added after water control (Fig. 1a). After washing with SDS, to potentially wash away non-covalently bound BSA-FITC, the fluorescent signal was $40 \pm 15$ fold higher ($p < 0.001$) in the EDC-NHS group compared to BSA only. Together, these results strongly suggest that covalent coupling of BSA-FITC was achieved.

Next, we used this validated strategy to couple bFGF to microfibrillar substrates. As opposed to bFGF in solution, cell response to covalently coupled bFGF has not been widely studied. In an attempt to find the right concentration range, we coupled three different amounts of bFGF to microfibrillar substrates. As we could not readily measure the amount of bFGF bound to the microfibrillar substrates, we measured the bFGF that was left over in solution after coupling. After incubation with the bFGF solution to couple bFGF to the microfibrillar substrates, we measured the bFGF that was left over in this solution (thus not coupled to the microfibrillar scaffolds) with ELISA. (Fig. 1b). Without the addition of EDC-NHS, ~30% of the original concentration of bFGF was left over in the bFGF solution with which the scaffold was incubated, meaning that ~70% of the bFGF adhered aspecifically to the scaffolds at all three concentrations. With the addition of EDC-NHS, ~98% of bFGF was bound to the scaffolds, aspecifically or covalently. Before cell culture, scaffolds were thoroughly washed in water and PBS, to attempt to wash away aspecifically bound bFGF.

To test whether the bound bFGF was still functional, proliferation of hMSCs cultured on the microfibrillar substrates was assessed after 7 days (Fig. 1c). Interestingly, the hMSCs did not respond to either bFGF bound to the microfibrillar substrates, or bFGF in solution. In 2D tissue culture plastic, hMSCs did increase proliferation over 7 days in response to bFGF in solution, displaying $45 \pm 11\%$ more DNA, demonstrating that the microfibrillar environment influenced the hMSC's response to bFGF (Supplementary Fig. 2).

Fibroblasts are particularly well studied for their increase in proliferation in response to bFGF. To test whether this lack of response to bFGF when cultured on microfibrillar substrates was specific to hMSCs, human dermal fibroblasts were cultured for 7 days on the microfibrillar substrates. On non-functionalized scaffolds, $77 \pm 20\%$ more ($p < 0.0001$) DNA was found after 7 days of culture in the presence of bFGF in the medium (Fig. 1d). On the 1000 ng covalently coupled bFGF scaffolds, $50 \pm 13\%$ more ($p < 0.01$) DNA was found compared to non-functionalized scaffolds, showing that the covalently bound bFGF was still functional.

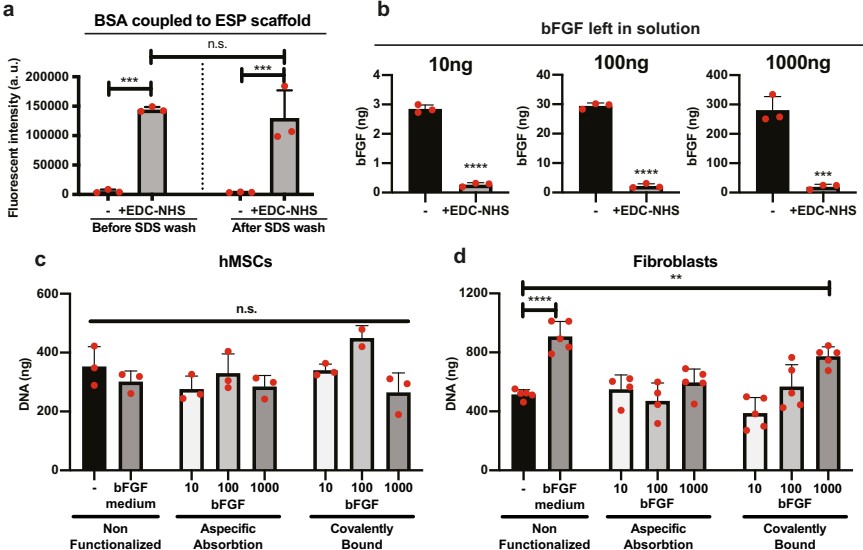

**Fig. 1 Functional coupling of bFGF to microfibrillar substrates. a** Fluorescent model protein BSA coupled to microfibrillar substrates using EDC-NHS, or water (−). Right bars are the same scaffolds after overnight wash with 1% (w/v) SDS in water. $n = 3$ scaffolds per condition. **b** Measurement of bFGF left in solution by ELISA after functionalization of 10, 100, or 1000 ng bFGF per scaffold, using EDC-NHS + bFGF, or water+bFGF(−). $n = 3$ scaffolds per condition. **c, d** DNA quantification of hMSCs (**c**) or human dermal fibroblasts (**d**) cultured on microfibrillar substrates functionalized with 10, 100, or 1000 ng bFGF per scaffold using bFGF+EDC-NHS (covalently bound, right 3 bars), bFGF + water (aspecific absorption, middle 3 bars), or non-functionalized scaffolds (left two bars). Cells were cultured in basic medium, or in medium supplemented with 10 ng/ml bFGF (bFGF medium condition). $n = 3$ scaffolds per condition for **c**, and $n = 5$ scaffolds per condition for **d**. **a, c, d** One-way ANOVA with Tukey's post-hoc test. **b** Student's $t$-test. **a–d**, n.s. $p > 0.05$; $**p < 0.01$; $***p < 0.001$; $****p < 0.0001$. Error bars indicate mean ± SD. Individual data points as red dots.

Heparin is known to bind and stabilize bFGF and increase efficacy[30]. To covalently couple heparin to the microfibrillar substrates, PEG-NH$_2$ was incorporated into the electrospinning polymer solution to introduce amino groups on the surface of the microfibrillar substrates. The carboxyl groups of heparin were then bound to the microfibrillar substrates by EDC-NHS chemistry (Supplementary Fig. 3a). bFGF was then bound to the heparin-functionalized scaffolds by overnight incubation. As the heparin interfered with the bFGF ELISA, the amount of absorbed bFGF could not be measured. After 7 days of culture, no differences were observed between hMSCs cultured on heparin +bFGF scaffolds and the heparin only- or non-functionalized PEG-NH$_2$ scaffolds (Supplementary Fig. 3b). This further demonstrates that hMSCs do not respond to bFGF on microfibrillar substrates, also not when bFGF is bound to heparin.

Together, these results show that the covalently coupled bFGF was still functional and supports proliferation in fibroblasts, but that hMSCs do not respond to bFGF when cultured on microfibrillar substrates.

**Reduced FGFR1 expression on microfibrillar substrates in hMSCs, but not fibroblasts**. To test why fibroblasts and hMSCs responded differently to bFGF tethered to microfibrillar substrates, we analyzed FGF receptor 1 (FGFR1) expression of hMSCs and fibroblasts cultured on TCP, as well as 2D films and microfibrillar substrates made of the same material. Interestingly, when cultured on microfibrillar substrates, hMSCs expressed 87 ± 5% less ($p < 0.01$) FGFR1 than when cultured on TCP (Fig. 2a). On films, hMSCs displayed 67 ± 7% less ($p < 0.01$) FGFR1 expression than on TCP, showing that part of the reduction of FGFR1 expression on microfibrillar substrates comes from the material properties. However, on microfibrillar substrates the FGFR1 expression was still 60 ± 16% lower ($p < 0.05$) than on films, showing that regardless of material properties, the microfibrillar environment influenced FGFR1 expression.

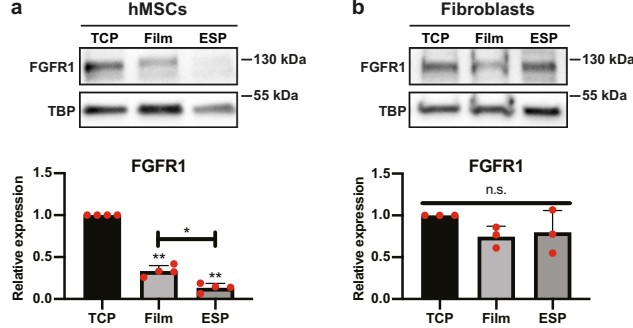

**Fig. 2 FGFR1 expression of hMSCs and fibroblasts on TCP, films and microfibrillar substrates. a, b** Western blot of FGFR1 and TBP (as loading control) of hMSCs (**a**) or human dermal fibroblasts (**b**) on TCP, films or microfibrillar substrates (ESP). Graphs depict quantification of western blots of FGFR1/TBP from 4 (**a**), or 3 (**b**) independent experiments, normalized to TCP. **a, b** Repeated measures ANOVA with Tukey's post-hoc test. Stars above bars indicate significance compared to TCP. n.s. $p > 0.05$; $*p < 0.05$; $**p < 0.01$. Error pars indicate mean ± SD. Individual data points as red dots.

Fibroblasts, however, did not display a difference in FGFR1 expression between the different culture substrates (Fig. 2b). The reduced FGFR1 expression of hMSCs on microfibrillar substrates, and the high FGFR1 expression of fibroblasts on microfibrillar substrates, potentially explains the difference in bFGF response of hMSCs and fibroblasts on microfibrillar substrates. Full unedited blots can be found in Supplementary Fig. 4.

**hMSCs, but not fibroblasts, display fewer focal adhesions on microfibrillar substrates**. To understand why hMSCs reduced FGFR1 expression on microfibrillar substrates, but fibroblasts did not, we investigated the difference in adhesion to the different substrates in hMSCs and fibroblasts by looking at focal adhesions.

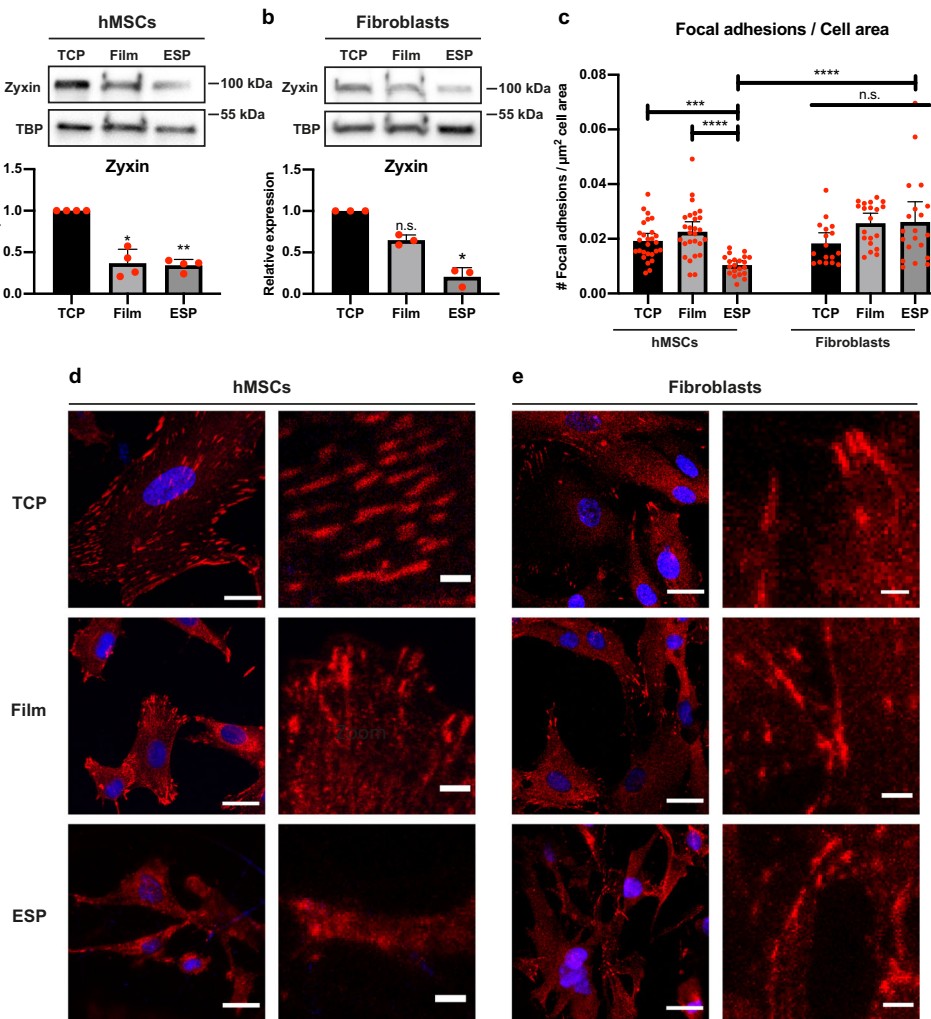

**Fig. 3 Zyxin expression and focal adhesion analysis of hMSCs and fibroblasts on TCP, films, and microfibrillar substrates. a, b** Western blot of zyxin and TBP (as loading control) of hMSCs (**a**) or human dermal fibroblasts (**b**) on TCP, films or microfibrillar substrates (ESP). Graphs depict quantification of western blots of zyxin/TBP from 4 (**a**), or 3 (**b**) independent experiments, normalized to TCP. Stars indicate significance compared to TCP. Repeated measures ANOVA with post-hoc test. Error bars indicate mean ± SD. **c** quantification of number of zyxin positive focal adhesions per μm² cell area of hMSCs or human dermal fibroblasts grown on TCP, films, or microfibrillar substrates (ESP). $n = 17$–27 cells, quantified in 5–10 different images from biological triplicates. Kruskal–Wallis test with post-hoc test. Error bars indicate mean ± 95% CI. **a–c** n.s. $p > 0.05$; *$p < 0.05$; **$p < 0.01$; ****$p < 0.0001$. Individual data points as red dots. **d, e** Representative images of hMSCs (**d**) or human dermal fibroblasts (**e**) stained for zyxin (red) and nuclei (blue). Right panels represent a ×5 magnification of the respective left panel. Scalebars represent 25 μm (left panels) and 4 μm (right panels).

The expression of zyxin, an important focal adhesion protein, was reduced in both hMSCs and fibroblasts, respectively by 66 ± 7% ($p < 0.01$) and 79 ± 11% ($p < 0.05$) compared to TCP (Fig. 3a, b). Paxillin expression, another well studied focal adhesion protein, was significantly reduced in both hMSCs and fibroblasts on microfibrillar substrates, compared to TCP; respectively 73 ± 5% ($p < 0.01$) and 65 ± 8% ($p < 0.05$; Supplementary Fig. 5a, b). On films, hMSCs also displayed reduced zyxin and paxillin expression, respectively 63 ± 17% and 41 ± 11% compared to TCP. Fibroblasts did not show a significant difference in zyxin or paxillin expression on films, compared to TCP. Full unedited blots can be found in Supplementary Figs. 6 and 7.

When looking at the formation of zyxin positive focal adhesions, a reduction of 46 ± 18% ($p < 0.01$) of focal adhesions per cell area was observed when hMSCs were cultured on microfibrillar substrates, compared to TCP (Fig. 3c, d). When compared to films, hMSCs on microfibrillar substrates displayed 54 ± 16% ($p < 0.0001$) less zyxin positive focal adhesions per cell area. Interestingly, no significant difference was found between

fibroblasts cultured on the different substrates (Fig. 3c, e). Indeed, when compared to fibroblasts grown on microfibrillar substrates, hMSCs on microfibrillar substrates displayed 60 ± 14% ($p < 0.0001$) fewer focal adhesions per cell area. The same trend was observed for paxillin positive focal adhesions, where hMSCs displayed far fewer paxillin positive focal adhesions on micro-fibrillar substrates than on films or TCP, while fibroblasts contained many paxillin positive focal adhesions on all three substrates (Supplementary Fig. 5c, d).

These results demonstrate that the microfibrillar environment changes focal adhesion formation in hMSCs, but not in fibroblasts. This shows that hMSCs adhere differently to the microfibrillar substrates than fibroblasts, potentially explaining the difference in FGFR1 expression.

As the lower FGFR1 expression correlated with fewer focal adhesions of hMSCs on microfibrillar substrates, we knocked down paxillin and zyxin in hMSCs cultured on TCP. Interestingly, neither paxillin nor zyxin depletion resulted in a change in FGFR1 expression, demonstrating that the differential expression

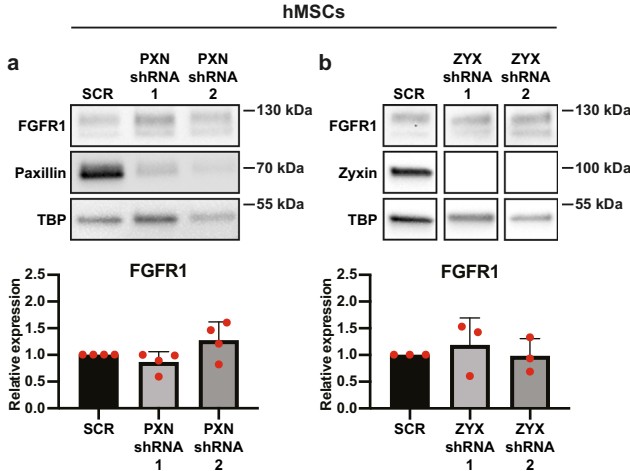

**Fig. 4 No role of paxillin or zyxin in regulation of FGFR1. a, b** Western blot of FGFR1, paxillin, zyxin, and TBP (as loading control) of hMSCs transduced with PXN-shRNA (**a**) or ZYX-shRNA (**b**) cultured on TCP. Graphs depict quantification of western blots of FGFR1/TBP from four biological replica's, normalized to TCP. **a, b** Repeated measures ANOVA with Tukey's post-hoc test. Error bars indicate mean ± SD. Individual data points as red dots.

of these proteins by hMSCs on microfibrillar substrates is not the reason for the reduced FGFR1 expression (Fig. 4a, b). Full unedited blots can be found in Supplementary Fig. 8.

**SRF and MRTF-A correlate with FGFR1 expression.** Even though focal adhesions did not influence FGFR1 expression, the reduction in focal adhesions of hMSCs on microfibrillar substrates suggests a difference in mechanosensitive signaling. An important mechanosensitive pathway is the MRTF/SRF pathway. MRTF translocates to the nucleus when actin is incorporated into actin filaments and globular actin is low, where it activates SRF to transcribe target genes[24]. To investigate this pathway, we looked at the expression of SRF. Indeed, compared to TCP, the SRF expression was $73 \pm 11\%$ ($p < 0.05$) lower on films and $92 \pm 6\%$ ($p < 0.01$) lower on microfibrillar substrates in hMSCs. Compared to films, SRF expression was $70 \pm 22\%$ ($p < 0.05$) lower on microfibrillar substrates (Fig. 5a). For fibroblasts, the expression of SRF was $25 \pm 8\%$ ($p > 0.05$) and $50 \pm 20\%$ ($p > 0.05$) lower on films and microfibrillar substrates respectively, but this difference was not statistically significant (Fig. 5b). Full unedited blots can be found in Supplementary Fig. 9.

To further investigate the MRTF/SRF pathway, we looked at the localization of MRTF-A in hMSCs and fibroblasts, on TCP, films and microfibrillar substrates. In hMSCs on TCP, MRTF-A was located almost exclusively in the nucleus (Fig. 5c). On films, MRTF-A was found in the cytoplasm, with nuclear to cytoplasmic ratio of MRTF-A $93 \pm 2\%$ less ($p < 0.0001$) than on TCP (Fig. 5d). On microfibrillar substrates, MRTF-A was located in the nucleus and in the cytoplasm, with $45 \pm 13\%$ less ($p < 0.05$) nuclear translocation than on TCP. Similar to hMSCs, MRTF-A was located in the nucleus in fibroblasts on TCP. (Fig. 5e). On films and microfibrillar surfaces, MRTF-A was located both in the cytoplasm and in the nucleus, with $44 \pm 14\%$ ($p < 0.001$) and $47 \pm 14\%$ ($p < 0.001$) less nuclear translocation than on TCP, respectively (Fig. 5f).

Together with the SRF expression, these results suggest activity of the MRTF/SRF pathway in fibroblasts on all substrates and of hMSCs on TCP, but not of hMSCs on films or microfibrillar substrates. The activity of the MRTF/SRF pathway correlates with the reduced FGFR1 expression of hMSCs on films or microfibrillar substrates.

**Actin-myosin and MRTF/SRF pathway regulate FGFR1 expression.** To investigate the role of the MRTF/SRF pathway in the regulation of FGFR1 in hMSCs, we inhibited the pathway using CCG203971[31,32]. Indeed, in both hMSCs and fibroblasts, inhibition of the MRTF/SRF pathway reduced FGFR1 expression by $60 \pm 7\%$ ($p < 0.01$) and $62 \pm 3\%$ ($p < 0.01$), respectively (Fig. 6a, c). This shows that MRTF/SRF directly or indirectly regulates FGFR1 expression in both hMSCs and fibroblasts. We observed a strong decrease in SRF expression in hMSCs on microfibrillar substrates (Fig. 5a), strongly suggesting that the reduced FGFR1 expression of hMSCs on microfibrillar substrates is due to a decrease in SRF expression. Fibroblasts maintained a high expression of SRF on microfibrillar substrates (Fig. 5b), which supports the high expression of FGFR1 on microfibrillar substrates.

When most actin monomers are assembled into filaments and globular actin is low, the MRTF/SRF pathway is activated. To determine the role of actin-myosin in the regulation of FGFR1, we treated hMSCs and fibroblasts with blebbistatin, a myosin inhibitor that greatly disrupts F-actin fibers. Expression of FGFR1 reduced $48 \pm 10\%$ ($p < 0.05$) in hMSCs and $42 \pm 13\%$ in fibroblasts (Fig. 6b, d). Together, this demonstrates that FGFR1 is regulated by the actin cytoskeleton and through the MRTF/SRF pathway. Full unedited blots can be found in Supplementary Fig. 10.

Another important mechanosensitive co-transcription factor is Yes activated protein 1 (YAP), entering the nucleus when a cell experiences high cellular tension[11]. To investigate if YAP plays a role in FGFR1 regulation, we knocked down YAP in hMSCs. No difference was observed in FGFR1 expression between YAP-knockdown and control-shRNA groups (Supplementary Fig. 11), demonstrating that YAP does not play a role in FGFR1 regulation in hMSCs. Full unedited blots can be found in Supplementary Fig. 12.

To further investigate the link between the MRTF/SRF pathway and the FGF pathway, we investigated the response to bFGF of hMSCs cultured with MRTF/SRF inhibitor. After 7 days of culture on TCP in the presence of bFGF and/or MRTF/SRF inhibitor, total DNA was analyzed. As expected, $36 \pm 9\%$ more DNA was found when bFGF was added to the medium, compared to basic medium (Fig. 6e). In the presence of MRTF/SRF inhibitor, $53 \pm 5\%$ less DNA was found than in basic medium. Interestingly, in the presence of MRTF/SRF inhibitor, hMSCs did not increase proliferation when bFGF was added. This shows that the MRTF/SRF pathway regulates the response to bFGF, in confirmation with the reduced FGFR1 expression.

Aberrant FGFR regulation in cancer cells has been linked to metastasis, tumor progression and a worse diagnosis. To test whether the MRTF/SRF pathway is also responsible for FGFR1 regulation in cancer cells, we treated the osteosarcoma cell line MG63 with the MRTF/SRF inhibitor. Similar to hMSCs and fibroblasts, FGFR1 expression was reduced by $60 \pm 10\%$ ($p < 0.05$) when MRTF/SRF was also inhibited in MG63 cells, which were used as a further cell source to investigate the correlation between FGFR1 and MRTF/SRF pathway (Supplementary Fig. 13). Full unedited blots can be found in Supplementary Fig. 14. MRTF/SRF inhibition decreased FGFR1 expression in three different human cell types, suggesting that the MRTF/SRF pathway is a univocal regulator of the FGFR1 pathway.

## Discussion

Here, we functionalized 300PEOT45PBT55 microfibrillar substrates by coupling bFGF to the surface. The covalent binding of bFGF to the microfibrillar substrates made of other polymers has been shown before to retain the growth factor bioactivity[29,33]. Similarly, the covalently coupled bFGF was still active on our

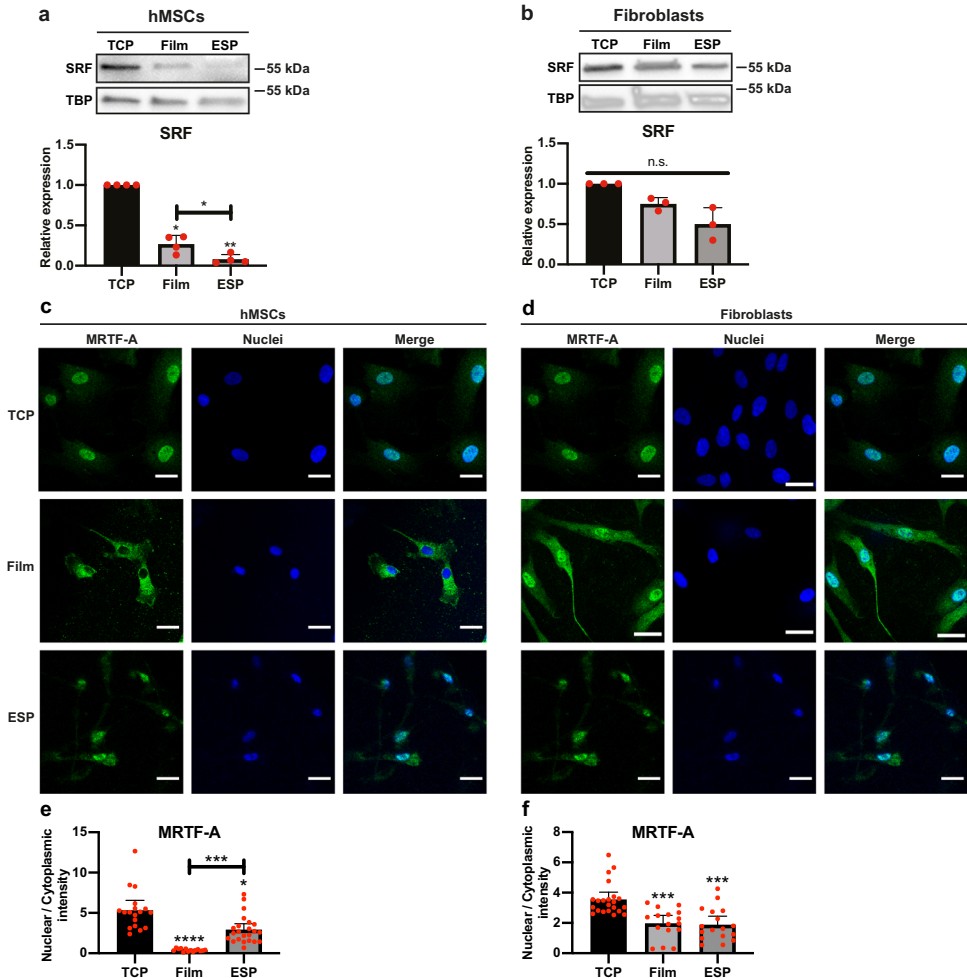

**Fig. 5 Reduced nuclear MRTF-A on films and microfibrillar substrates. a**, **b** Western blot of SRF and TBP (as loading control) of hMSCs (**a**) or human dermal fibroblasts (**b**) cultured on TCP, films or microfibrillar substrates (ESP). Graphs depict quantification of western blots of SRF/TBP from 4 (**a**) or 3 (**b**) independent experiments, normalized to TCP. Repeated measures ANOVA with Tukey's post-hoc test. Error bars indicate mean ± SD. **c**, **d** hMSCs (**c**) and fibroblasts (**d**) were cultured for 7 days on TCP, films, or microfibrillar substrates (ESP) and stained for MRTF-A (green) and nuclei (blue). Scalebars represent 30 μm. **e**, **f** Quantification of MRTF-A nuclear localization in hMSCs (**e**) and fibroblasts (**f**). MRTF-A staining intensity normalized to cytoplasmic staining intensity. $n = 14$-23. Kruskal–Wallis test. Error bars indicate mean ± 95% CI. **a**, **b**, **e**, **f**, n.s. $p > 0.05$; *$p < 0.05$; ***$p < 0.001$; ****$p < 0.0001$. Stars above bars indicate significance compared to TCP. Individual data points as red dots.

microfibrillar substrates and could be used as a method to increase fibroblast proliferation on microfibrillar substrates. This observation suggests that other cell types could fall somewhere in the spectrum between responsive and non-responsive when cultured on these bFGF functionalized microfibrillar scaffolds. This could be useful for in vivo approaches, but it can also be used as a cell culture substrate in vitro. bFGF is highly unstable in solution and covalent binding to a surface has been shown to increase its stability[29]. Further investigation and characterization of the functionalized scaffolds would be useful for such uses. For example, the precise amount of covalently bound bFGF, the amount and effect of the potentially left over absorbed bFGF, and the stability of the covalently bound bFGF would be interesting parameters at the material interface to study. Also, in the highest concentration of coupled bFGF (1000 ng) almost all bFGF bound to the microfibrillar substrates (Fig. 1b), suggesting that saturation has not yet been reached. As the fibroblasts only responded to the 1000 ng bFGF substrates, higher concentrations might further increase proliferation. It would be interesting to investigate the optimal bFGF density to induce cell proliferation on microfibrillar substrates, while keeping in mind

that this might differ for different cell types. By mapping cell size, cell migration and contact area with the microfibers, one could also approximate the required quantity of bound bFGF molecules with which a cell needs to come in contact to induce cell proliferation. This could be supplemented with data from different concentrations of soluble bFGF to determine whether there is a difference in cell response to bound and soluble bFGF. This research could lead to an in-depth understanding of how scaffold properties such as fiber diameter, fiber spacing, and ability to be remodeled influence the amount of contact that cells have with bound bFGF molecules, and thus how these factors can influence proliferation. Other factors such as cell size and migration could then also be investigated for their role in growth factor-induced proliferation. Together, such an in-depth understanding of how cells interact with the surrounding environment could greatly aid both our fundamental understanding of cell behavior in 3D and the smarter design of regenerative medicine scaffolds.

Unlike fibroblasts, hMSCs did not increase proliferation in response to bFGF (in solution or covalently bound) on microfibrillar substrates. We found that this was due to reduced SRF

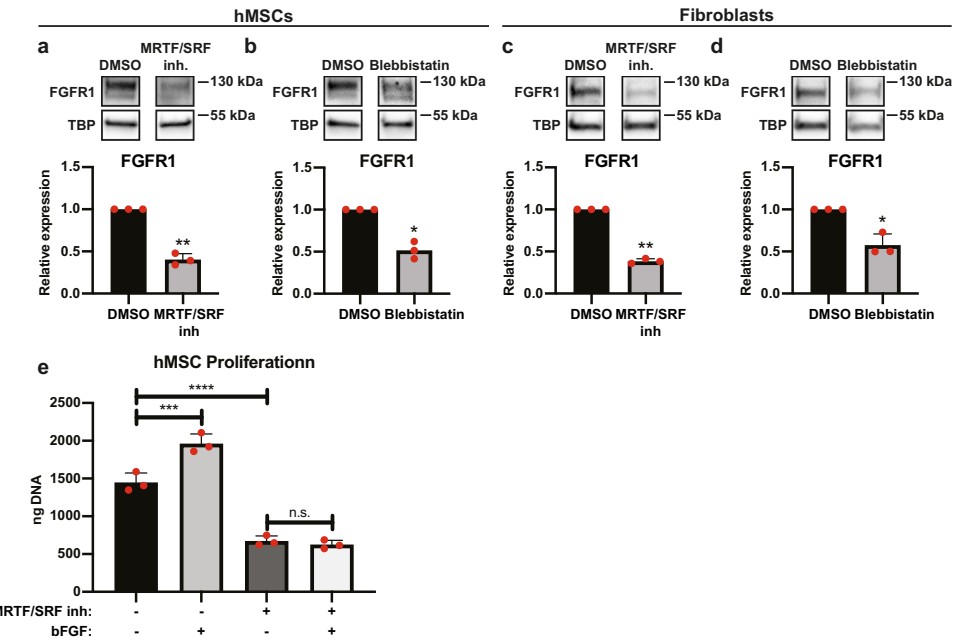

**Fig. 6 Actin-myosin and MRTF/SRF inhibitors change FGFR1 expression in hMSCs and fibroblasts. a–d** Western blot of FGFR1 and TBP (as loading control) of hMSCs (**a**, **b**) or human dermal fibroblasts (**c**, **d**), cultured on TCP and treated with MRTF/SRF inhibitor CCG203971 (**a**, **c**) or blebbistatin (**b**, **d**). Graphs depict quantification of western blots of FGFR1/TBP from three biological replica's, normalized to TCP. **e** DNA quantification of hMSCs cultured for 7 days on TCP in the presence of MRTF/SRF inhibitor and/or 10 ng/ml bFGF. $n = 3$ for each condition. One-way ANOVA with Tukey's post-hoc test. n.s. $p > 0.05$; ***$p < 0.001$; ****$p < 0.0001$; Error bars indicate mean ± SD. **a–d**, **f**, ratio-paired $t$-test. *$p < 0.05$; **$p < 0.01$. Error bars indicate mean ± SD. Individual data points as red dots.

expression, which caused decreased FGFR1 expression. SRF expression is known to be regulated by itself through a positive feedback loop[34]. The observed difference in SRF expression between TCP, films, and microfibrillar substrates highlight the difference in SRF activity on the different substrates. The positive feedback loop can exaggerate the differences in SRF expression, but the origin of the initial difference in SRF expression remains unclear. The 300PEOT45PBT55 material itself also affected FGFR1 and SRF expression, as seen by reduced FGFR1 and SRF expression in hMSCs on films vs TCP. The microfibrillar substrates further decreased this expression on the same material, showing a direct effect of the microfibrillar environment on FGFR1 and SRF expression. MRTF-A was located in the nucleus of hMSCs and fibroblasts on TCP. Together with high SRF activity, this suggested that the MRTF/SRF pathway was active. Fibroblasts on films and microfibrillar substrates also displayed nuclear MRTF-A, although less than on TCP. As SRF expression did not significantly change on the different substrates, this could explain the high FGFR1 expression and responsiveness to bFGF on microfibrillar substrates. hMSCs on films did not show nuclear localization of MRTF-A, which together with the low SRF expression suggests that the pathway is inactive, explaining the low FGFR1 expression. On microfibrillar surfaces, MRTF-A was located partly in the nucleus, although less than on TCP. Even though MRTF-A was located in the nuclei, the low SRF expression of hMSCs on microfibrillar substrates could prevent active transcription of the FGFR1 gene, or of genes that (indirectly) regulate FGFR1. We cannot exclude the possibility that MRTF-A does not play a role in FGFR1 regulation, although almost all SRF target genes are regulated by MRTF[35]. As with FGFR1 and SRF expression, MRTF-A localization was also affected by the 300PEOT45PBT55 material as well as directly by the microfibrillar environment. It remains to be investigated how the microfibrillar environment and which of the material properties affected the expression of these proteins.

Through actin-myosin inhibition by blebbistatin, we found that FGFR1 expression is reduced with less actin-myosin tension. The MRTF/SRF pathway is dependent on the actin cytoskeleton, but also plays a role in shaping the actin network[36]. We did not investigate whether the effect of actin-myosin inhibition went through MRTF/SRF, or vise-versa. It is possible that no clear cause and effect between these two players exists, because there is a positive feedback look between the two. MRTF/SRF activity increases stress fiber formation, thereby also increasing MRTF nuclear localization and increasing MRTF/SRF activity[36].

hMSCs grown on microfibrillar substrates displayed fewer focal adhesions than on films or TCP. In contrast, fibroblasts formed similar numbers of focal adhesions per cell area on microfibrillar substrates as on films or TCP. On TCP and films, the number of zyxin positive focal adhesions was the same between hMSCs and fibroblasts. Knockdown of either zyxin or paxillin did not affect FGFR1 expression. In contrast to paxillin, zyxin knockdown is known to diminish stress fibers[37–39]. While actin-myosin inhibition by blebbistatin did decrease FGFR1 expression, zyxin knockdown did not. Although fewer than normally, zyxin knockdown cells still form focal adhesions[25,40]. Our data suggests that the actin-myosin tension between these focal adhesions is still sufficient to maintain a higher FGFR1 expression, as full inhibition of actin-myosin by blebbistatin reduced FGFR1.

The reason for the difference between fibroblasts and hMSCs was not investigated here. Different cell types exhibit different cell spreading and traction forces in response to different substrate stiffnesses[41–43]. Indeed, the optimal stiffness for differentiation and proliferation defer per cell type[44,45]. We have previously shown that hMSCs experience the microfibrillar substrates used here as a soft substrate, demonstrated by fewer focal adhesions, less lamin A/C and less YAP nuclear translocation[25]. The difference in focal adhesion formation between hMSCs and fibroblasts observed here potentially derives from a different response to matrix stiffness. Perhaps fibroblasts are able to form focal

adhesions on softer substrates than hMSCs. A side by side comparison of hMSCs and fibroblasts on different stiffnesses has not yet been reported but could shed light on the differences observed here. It is also possible that besides a difference in focal adhesion formation, other phenotypical differences between hMSCs and fibroblasts play a role in FGFR1 regulation and their response to bFGF. We have shown that both hMSCs and fibroblasts regulate FGFR1 through MRTF/SRF and actin-myosin, but how MRTF/SRF and actin-myosin are regulated could differ significantly between the two cell types. Such potential differences have not yet been thoroughly investigated. Additionally, other proteins that are differently expressed between hMSCs and fibroblasts could also affect FGFR1 expression. A deeper investigation in different proteins that regulate FGFR1 and how they are regulated would aid in explaining the observed differences between hMSCs and fibroblasts. For example, DNA pull down of FGFR1 promotor regions could be used to identify novel transcription factors. The role of mechanosensitive pathways in FGFR1 regulation that we have shown here could then be used to identify how these novel transcription factors are regulated.

The regulation of FGFR1 expression is, however, poorly studied. With the inhibitors of MRTF/SRF and actin-myosin, we could show that these protein complexes greatly influence FGFR1 expression. It is of course possible that other factors also further influenced the expression of FGFR1. This could be due to downstream effects of the change in actin-myosin, or through co-activation of independent pathways. YAP knockdown has been shown to decrease FGFR1 expression in lung cancer cells[9] and neurospheres[10]. Also, integrin α6 has been shown to regulate FGFR1[10]. We found that YAP knockdown didn't alter FGFR1 expression in hMSCs, suggesting a different role for YAP in different cell types. YAP and integrin α6 regulation of FGFR1 does, however, hint at the mechanosensitive regulation of FGFR1, in accordance with what we have shown here. Other proven mechanisms of FGFR1 regulation include regulation by Pdx-1[46,47] and ZEB1[10]. Whether these proteins play a role in FGFR1 regulation on microfibrillar substrates has not been investigated here. The regulation of FGFR1 by MRTF/SRF and actin-myosin tension presented here adds to the sparse literature on FGFR1 regulation.

These findings might also give insight in tumor development, as aberrant FGFR1 regulation is important in a wide variety of cancers[48,49]. Using next generation sequencing to analyze 4853 tumors, Helsten et al.[48] found aberrations in FGFR in 7.1% of all tumors. In addition, increased expression of FGFR has been correlated with a bad prognosis, increased metastasis and tumor progression in a large variety of cancers[49–53]. Indeed, animal studies and clinical trials are currently ongoing to test the effects of FGFR inhibitors on cancer treatment, showing promising initial results[54–59]. Unraveling FGFR regulation could advance the understanding of tumor development and open up new therapeutic targets. Although only one experiment with an osteosarcoma cell line (MG63) is presented here, our study may open up new potential targets for FGFR1 regulation in cancer cells. Also, as an important regulator of proliferation in hMSCs[8] and other cell types[60,61], this can have implications for scaffold designs. We show here that the scaffold design itself, as well as material properties, can influence FGFR1 expression. Optimizing scaffold design to influence MRTF/SRF activity and FGFR1 expression could be crucial for tissue regeneration applications.

## Conclusion

Microfibrillar substrates were successfully functionalized with bFGF, which increased the proliferation of fibroblasts, but not of hMSCs. hMSCs, but not fibroblasts, reduced FGFR1 expression on microfibrillar substrates, explaining the lack of response to bFGF on microfibrillar substrates. Fibroblasts maintained a high expression of FGFR1 on microfibrillar substrates, explaining the difference in bFGF responsiveness between hMSCs and fibroblasts. hMSCs, but not fibroblasts, displayed fewer focal adhesions and expressed less SRF on microfibrillar substrates than on TCP or 2D film controls. In hMSCs and fibroblasts, the inhibition of actin-myosin interaction and MRTF/SRF activity decreased FGFR1 expression. In osteosarcoma MG63 cells, MRTF/SRF inhibition also led to decreased FGFR1 expression. Together, our data shows that hMSCs become irresponsive to bFGF on microfibrillar substrates because of a downregulation of SRF, which leads to a decrease in FGFR1. Fibroblasts maintain a high SRF and FGFR1 expression and remain responsive to bFGF on microfibrillar substrates.

## Methods

**Film and microfibrillar substrate production**. Random block co-polymer of poly (ethylene oxide terephthalate) (PEOT) and poly(butylene terephthalate) (PBT), with 300 Da PEG and PEOT/PBT ratio (w/w) of 55/45 (300PEOT55PBT45, acquired from PolyVation) was used to produce films and microfibrillar substrates. 300PEOT55/PBT45 granules were melted at 180 °C under slight pressure (~100 kg) in a circular 23 mm mold between two silicon wafers (Si-mat, Kaufering, Germany) to produce flat films. Films were punched out using a 22 mm punch to fit in a 12-well plate.

The electrospinning polymer solution was prepared by dissolving 20% (w/v) 300PEOT55PBT45 in 3:7 1,1,1,3,3,3,-Hexafluoro-2-propanol (HFIP):chloroform, overnight at room temperature under agitation. For heparin functionalization, 2% (w/v) poly(ethylene glycol) (PEG) with 2 NH$_2$ end-groups (Mw: 3350 kDa) (PEG-NH$_2$), was added to the polymer solution and mixed for 4 h before electrospinning.

ESP scaffolds (microfibrillar substrates) were produced on a slowly rotating (100 RPM) 19 cm diameter mandrel by electrospinning on a polyester mesh (FinishMat 6691 LL (40 gr/m2), generously provided by Lantor B.V.) with 12 mm holes, on top of aluminum foil. The following parameters were maintained: 15 cm working distance between needle and rotating mandrel, 1 ml/h flow rate, 23–25 °C and 40% relative humidity, a needle charge between 10 and 15 kV and collector charge between −2 and −5 kV. Individual ESP scaffolds were punched out with a diameter of 15 mm over the 12 mm holes in the polyester mesh and removed from the aluminum foil. This resulted in 15 mm ESP scaffolds with a 12-mm diameter surface for cell culture and a 1.5-mm polyester ring around it to improve handleability. Using this method, up to 100 microfibrillar substrates were produced under exactly equal parameters.

Before cell culture, microfibrillar substrates and films were sterilized in 70% ethanol for 15 min and dried at room temperature until visually dry. The 1.5-mm polyester ring was covered with a rubber 15 mm outer- and 12 mm inner-diameter O-ring (Eriks) to keep the scaffolds from floating in tissue culture well plates.

**Functionalization of microfibrillar substrates with BSA or bFGF**. Before coupling of bovine serum albumin (BSA)-FITC conjugate (ThermoFisher Scientific) or basic fibroblast growth factor (bFGF) (Neuromics), ethanol sterilized microfibrillar substrates were incubated in 0.5 M NaOH for 30 min at room temperature to open the ester bond of the 300PEOT55PBT45 polymer. Scaffolds were thoroughly washed five times with water and then incubated with 4 mg/ml N-(3-Dimethyla-minopropyl)-N′-ethylcarbodiimide hydrochloride (EDC) (Sigma-Aldrich) and 10 mg/ml N-hydroxysuccinimide (NHS) (Sigma-Aldrich) in milliQ water, or in milliQ water only, without EDC-NHS, as negative control, for 30 min at room temperature on a rocking plate. EDC-NHS solution was removed and 500 µl of 1 µg/ml BSA or 20, 200, or 2000 ng/ml bFGF (thus in total 10, 100, or 1000 ng bFGF) in water was added to the scaffolds in a 24-well-plate well and incubated overnight at 4 °C on a rocking plate.

For BSA-FITC functionalization, the following day scaffolds were washed five times with water and scaffold fluorescence was measured in the fluorescein channel on a Clariostar plate reader (BMG Labtech). For sodium dodecyl sulfate (SDS) wash, 1% (w/v) in water was added to the functionalized scaffolds and incubated under agitation at room temperature overnight. The following day, scaffolds were thoroughly washed five times with water and measured on the plate reader as described before.

For bFGF functionalized scaffolds, the bFGF solution with which the scaffolds were incubated was harvested, before any washes, to be analyzed by bFGF ELISA. The scaffolds were then washed five times with water, once with PBS and once with medium. For the bFGF scaffolds, all solutions were sterilized by filtration through at 0.2 µm filter. As we could not directly measure bFGF on the scaffolds like we measured the fluorescent BSA-FITC, we measured the bFGF that did not bind to the scaffolds. The amount of unbound bFGF was measured by quantifying the harvested bFGF solutions with which the scaffolds were incubated, using a bFGF ELISA Kit (Abcam), according to manufacturer's protocol. In this way, we

quantified the amount of bFGF that was left over in solution, and thus did not bind to the scaffolds. This measurement was then used to compare scaffolds with and without EDC/NHS and to estimate the amount of bFGF that bound to the scaffolds.

**Heparin functionalization of microfibrillar substrates**. In all, 1.5 mg/ml heparin sodium salt from porcine intestinal mucosa (Sigma-Aldrich) was mixed with 4 mg/ml EDC and 10 mg/ml NHS in water (or water only, without EDC-NHS, as negative control) and directly added to the 300PEOT55PBT45 + PEG-NH$_2$ microfibrillar substrates and incubated overnight at 4 °C.

To measure bound heparin, scaffolds were washed five times with milliQ water and stained for 30 min with an alcian blue staining solution (0.1% alcian blue, 10% ethanol, 0.1% acetic acid, 0.03 M MgCl$_2$ in water (all Sigma-Aldrich)). Scaffolds were washed once with MQ water and incubated for 30 min at room temperature in destaining solution (10% ethanol, 0.1% acetic acid, 0.03 M MgCl$_2$ in water). Scaffolds were washed again once with water and then incubated for 30 min in 1% SDS to extract the heparin-bound alcian blue from the scaffolds. The absorbance of this solution was measured in a Clariostar plate reader.

For cell culture, the heparin-functionalized scaffolds were washed five times with milliQ water and incubated overnight at 4 °C with 500 µl 2000 ng/ml bFGF. The following day scaffolds were washed five times with water, once with PBS and once with medium. All solutions were sterilized by filtration through at 0.2 µm filter.

**Cell culture**. Human dermal fibroblasts (Lonza) were expanded at 2000 cells/cm$^2$ in DMEM + Glutamax medium (Thermo Fisher Scientific) supplemented with 10% (V/V) fetal bovine serum (FBS) (Sigma-Aldrich).) Bone marrow derived hMSCs were isolated by Texas A&M Health Science Center[62]. Briefly, aspirated bone marrow was centrifuged to isolate mononuclear cells. The hMSCs were further expanded and tested for differentiation potential. hMSCs were received at passage 1 and were further expanded at 1000 cells/cm$^2$ in α-MEM + Glutamax medium (Thermo Fisher Scientific) supplemented with 10% FBS. MG63 cells (ATCC) were expanded at 5000 cells/cm$^2$ in DMEM + Glutamax+10% FBS medium. All cells were cultured in 37 °C in 5% CO$_2$ until reaching 70–80% confluency. Cells were trypsinised in 0.05% Trypsin and 0.53 mM EDTA (Thermo-Fisher Scientific) and hMSCs and fibroblasts were used for experiments at passage 5. MG63 cells were used at passage 90.

Unless otherwise stated, all experiments were harvested at day 7. For scaffold experiments, hMSCs and fibroblasts were cultured at 1000 cells/cm$^2$ in TCP and films, and 30.000 cells/microfibrillar substrate in growth medium with 100 U/ml penicillin-streptomycin. All other experiments were done in medium without penicillin-streptomycin. In bFGF medium conditions, 10 ng/ml bFGF was added to the medium.

For blebbistatin and MRTF/SRF inhibitor experiments, hMSCs and fibroblasts cells were seeded at 1000 cells/cm$^2$ on TCP and cultured for 6 days. MG63 cells were seeded at 5000 cells/cm$^2$ cultured for 2 days, because of a very high proliferation rate. After the initial culture period in growth medium, 100 µM blebbistatin (Sigma-Aldrich) in 0.2% DMSO in growth medium, or 12.2 µM MRTF/SRF inhibitor CCG203971 in 0.1% DMSO in growth medium, or respective DMSO control was added to the cells for 24 h.

To test the responsiveness of hMSCs to bFGF in the presence of MRTF/SRF inhibitor, hMSCs were seeded at 1000 cells/cm$^2$ in TCP and cultured for 7 days in 0.1% DMSO control, 0.1% DMSO + 10 ng/ml bFGF, 24.4 µM MRTF/SRF in 0.1% DMSO or 24.4 µM MRTF/SRF in 0.1% DMSO + 10 ng/ml bFGF, all in hMSC growth medium.

**DNA quantification**. To lyse cells for DNA quantification, cells were washed 2x with PBS and freeze-thawed dry twice before RLT lysis buffer (Qiagen) was added. Microfibrillar substrates were removed from the polyester ring after the last PBS wash. Samples were then freeze-thawed 3x in lysis buffer. TCP plates and films were scraped with a cell scraper after the first freeze-thaw in lysis buffer. Microfibrillar substrates were left in lysis buffer. Samples were diluted 100–400x, depending on expected number of cells per samples, in Tris-EDTA buffer (10 mM Tris-HCl, 1 mM EDTA, pH 7.5) and λ-DNA standard was made in the same final solution (0.25-1% RLT in Tris-EDTA buffer). Pico green assay (ThermoFisher Scientific) was then used to quantify DNA, according to the manufacturer's protocol.

**Protein isolation and western blot**. Protein was isolated in a custom lysis buffer to allow for the detection of membrane proteins with western blot. Other buffers, such as RIPA buffer, were tried for FGFR1 western blot without success. The buffer consisted of 150 mM NaCl, 0.5% sodium deoxycholate, 1% SDS, 1% NP-40, and 50 mM Tris-HCl in water, set to pH 7.4. The buffer was supplemented with cOmplete™ Mini EDTA-free Protease Inhibitor Cocktail (Sigma-Aldrich).

Samples were washed in cold PBS twice before lysis. Microfibrillar substrates were removed from the supporting polyester ring. To get sufficient proteins, 6–12 films or 15–20 microfibrillar substrates were combined in 300–400 µl for a single protein isolate. Experiments were repeated 3 or 4 times to obtain sufficient

replicates. In total, 6 or 10 cm dishes were used for TCP samples. TCP and film conditions were scraped in lysis buffer with cell scrapers. Microfibrillar substrates were submerged in lysis buffer and incubated for ~30 min in lysis buffer because the scaffolds were removed from the protein isolate. Samples were not spun down to maintain potentially non-dissolved membrane proteins in solution.

Pierce BCA protein assay kit (Thermo Fisher Scientific) was used to quantify total protein concentration. 20 µg protein was incubated in 10% 2-Mercaptoethanol (Sigma-Aldrich) in laemmli loading buffer (Bio-Rad) for 37 °C for 20 min for FGFR1 western blots and at 95 °C for 5 min for all other western blots. Samples were loaded into 4–15% polyacrylamide gels (Bio-Rad) and blotted to 0.45 µm PVDF membranes (Bio-Rad) using semi-dry transfer. Membranes were blocked in 5% (w/v) fat free milk (Bio-Rad) in TBS + 0.05% (v/v) tween-20 (Sigma-Aldrich) for 1 h, except for SRF western blots, which had to be blocked in 2% (w/v) BSA (VWR) + 0.05% tween-20 in PBS to work. Primary antibodies were incubated in their respective blocking buffer overnight at 4 °C. All antibodies were ordered from Abcam: FGFR1: ab76464 1/500; Paxillin: ab32084 1/1000; Zyxin: ab58210 1/1000; YAP1: ab52771 1/1000; SRF: ab53147 1/250; and TBP: ab51841 1/1000. Blots were incubated the following day with 0.33 µg/ml Goat-anti-rabbit or -mouse HRP (Bio-Rad) in blocking buffer for 1 h at room temperature. To visualize the protein bands, blots were incubated with Clarity Western ECL (Bio-Rad) for 1–5 min right before imaging.

**Immunofluorescence and imaging**. Cells were fixed with 3.6% (v/v) paraformaldehyde (Sigma-Aldrich) in PBS for 20 min at room temperature. To block and permeabilize, fixed cells were incubated in 2% (w/v) BSA + 0.1% (v/v) triton X (VWR) in PBS. Zyxin or paxillin (Abcam, ab58210 and ab32084, respectively, both 1/1000) were incubated in 2% (w/v) BSA + 0.05% (v/v) tween-20 in PBS overnight at 4 °C. The following day, 1/1000 Goat-anti-mouse Alexa Fluor 568 or Goat-anti-rabbit Alexa Fluor 488 was incubated overnight at 4 °C in 2% (w/v) BSA + 0.05% (v/v) tween-20 in PBS. The next day, samples were stained with DAPI (Sigma-Aldrich, 0.14 µg/ml in PBS + 0.05% tween-20) to stain nuclei. Images were taken on a confocal microscope.

Focal adhesions were quantified manually by counting the number of focal adhesions per cell using Fiji. Between 17 and 27 cells were counted per condition, from 5 to 10 different images from biological triplicates. Cell area was measured by manually outlining the cells and measuring surface area using Fiji. The number of focal adhesions was normalized to the cell area.

**Lentiviral production and transduction**. To produce lentiviral particles, human embryonic kidney 293FT (HEK) cells were seeded at 60,000 cells/cm$^2$ in DMEM + Glutamax+10% FBS. Cells were transfected with pMDLg pRRE, pMD2.G, pRSV Rev (Addgene) and one of the pLKO.1 shRNA plasmids using 5:1 lipofectamine 2000 (Thermo Fisher Scientific):DNA (v/w) 24 h after seeding. The following TRC pLKO.1 constructs (Dharmacon) were used: ZYX: TRCN0000074204 and TRCN0000074205; PXN: TRCN0000123134 and TRCN0000123136; YAP1: TRCN0000107265 and TRCN0000107266; and non-targeting shRNA control (RHS6848). Medium was changed 16 h post-transfection to hMSC growth medium. Lentivirus was harvested and filtered through a 0.45-µm filter 24 h and 48 h after the change to hMSC growth medium.

In all, 24 h after thawing at 1000 cells/cm$^2$, hMSCs were transduced with the lentiviral medium for 16 h. Medium was replaced with growth medium the following day. 48–72 h post transduction, medium was replaced with growth medium + 2 µg/ml puromycin for 72 h. A total of 9–10 days after thawing, hMSCs were passaged and seeded at 1000 cells/cm$^2$ on TCP for 7 days in growth medium before protein harvest.

**Statistics and reproducibility**. The statistical tests and number of biological replicates and/or experiments are stated in the figure subtexts. Each experiment used at least three biological replicas. Cells selected for quantification of focal adhesions were selected randomly. Films and electrospun scaffolds were also randomly assigned to different experimental groups. Shapiro–Wilk test was used to test for normal distribution of each experimental group before further statistical analysis. To test for significance of absolute differences in experiments with multiple comparisons between groups, a One-way ANOVA with Tukey's post hoc was performed. For relative differences between multiple experimental groups, log values were used for repeated measures ANOVA, with Tukey's post-hoc test. For experiments with a single comparison, two-tailed Student's $t$ test was used for absolute differences, and ratio–paired $t$-test for relative differences. Significance was set at $p < 0.05$. Statistical analysis was done using Graphpad Prism 8.

**Reporting summary**. Further information on research design is available in the Nature Research Reporting Summary linked to this article.

## Data availability

All raw data is stored and securely backed-up and available upon request. All data underlying the graphs is available as Supplementary Data. Other data are available on reasonable request from the corresponding author.

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

## Acknowledgements

We would like to thank Matt Baker and Paul Wieringa for the valuable discussions about the functionalization strategy of the electrospun scaffolds. We are grateful to the European Research Council starting grant "Cell Hybridge" for financial support under the Horizon2020 framework program (Grant #637308). Some of the materials used in this work were provided by the Texas A&M Health Science Center College of Medicine Institute for Regenerative Medicine at Scott & White through a grant from NCRR of the NIH (Grant #P40RR017447).

## Author contributions

J.Z. designed and performed experiments, analyzed data and wrote the manuscript. S.R. designed and performed experiments and analyzed data. L.M. designed experiments, analyzed data, and wrote the manuscript.

## Competing interests

The authors declare no competing interests.
