## [Peer Review File · Communications Biology]

Reviewers' comments:

Reviewer #1 (Remarks to the Author):

The article 'Mechanosensitive regulation of FGFR1 through the MRTF-SRF pathway' report about the lack of hMSCs to respond to bFGF signaling when cultured on ESP scaffolds and conclude with the help of further experiments and comparison to fibroblast cells that this lack of bFGF signaling is caused by a mechanosensitive downregulation of FGFR1 expression via the MRTF/SRF pathway. The authors address an interesting topic that could be relevant for the development and improvement of techniques to artificially grow tissues and that would need more investigation. However, the presented results are not supporting the claims sufficiently to justify publication at the current stage and further experiments would be required. It would be helpful if the authors could structure the introduction and discussion better to clarify the motivation of the study (e.g. why using hMSCs in ESP scaffolds) and how the findings could help the wider community. The methods are well described, and statistical analysis was performed to a sufficient level (see comments).

Major comments:

1. The study does not provide any direct evidence for the claim that FGFR1 regulation is mechanosensitive through the MRTF-SRF pathway. The authors show that FGFR1 expression is reduced in hMSCs on ESP scaffolds and that the MRTF-SRF pathway is downregulated in hMSCs on ESP scaffolds. But there is no evidence that this is mediated via a mechanosensitive pathway. There are not measures of the mechanical properties of ESP scaffolds or films and no measures indicating a different acto-myosin contractility in hMSCs on ESP scaffolds or films. If it would be a mechanosensitive pathway, then the alteration of the mechanical properties of the environment should be sufficient to explain the difference in signaling, i.e. the authors would have to provide evidence, that the ESP films and scaffolds provide a mechanically different environment to hMSCs leading to changes in acto-myosin activity/ contractility, and hence changes in MRTF localization or SRF levels.

Otherwise put: if the regulation would solely be caused by the mechanical differences of ESP scaffolds, one would expect no or very little changes in FGFR1 expression between hMSCs on TCP and ESP films.

2. To link acto-myosin activity with MRTF regulated FGFR1 expression, the authors treat cells with blebbistatin and observe 40-50% reduction of FGFR1 expression in both, fibroblasts and hMSCs on TCP cultures. However, in order to link this reduction to the MRTF pathway, the authors should report the localization of MRTF (nucleus vs cytosolic) upon blebbistatin treatment. It would also be good to compare the results with other inhibitors, e.g. LatA, that captures g-actin or jasplakinolide that stabilizes actin filaments.

3. The data on MRTF localization and SRF levels as function of the substrate hMSCs are cultured on, is not very conclusive. The authors state in the discussion, that hMSCs on ESP scaffolds display high nuclear localization of MRTF, but SRF levels show a clear trend which would counter their earlier argument that the FGFR1 expression is regulated via MRTF. In addition, there is little discussion why the fibroblasts don't show a reduction in FGFR1 expression while displaying a reduction in SRF levels. It might help to quantify the relative localization and to move the images to the main figure.

4. Display of data and statistics: it would be clearer to use box plots (including data points) instead of the chosen bar diagrams to display the spread of the measures. In connection to that, since the majority of the data is obtained from a small number of samples ($n = 3-5$) and the variation of both read outs is high, it is advisable to state the relative changes in full numbers only (e.g. $87 \pm 6\%$ instead of $86.5 \pm 5.6\%$).

Minor comments:

Introduction:

- 5th Paragraph: ...receptors (coming from 4 FGFR genes, FGFR1-4); tyrosine kinase receptors that can activate a variety... The phrase does not make sense. There seems to be something missing in between.
- Same paragraph: the last two phrases can be combined.
- 6th Paragraph, first phrase: development

Results:

- 1st paragraph would fit better into the introduction
- 2nd paragraph: EDC-NHS chemistry: Please write out abbreviations when they are used first in the text
- ...A 26.4 ± 0.8 fold (p...) : it's clearer to use 'fold' instead of 'x', be consistent with the separators (it's altering between ',' and '.'), write directly the measured number and not ~27 fold (...).
- Better formulation: After washing with SDS to remove unbound BSA... (you might not remove unspecifically binding BSA with the wash)
- 3rd paragraph: ... aspecifically to the scaffolds at all three concentrations
- 4th paragraph: can you give an estimation of the surface density of bFGF for the different concentrations at which bFGF was coupled to the ESP scaffolds?
- Reference to figure is missing for the DNA increase between non-functional and bFGF coupled ESP scaffolds.
- It's not clear why the tests with the heparin-coupled bFGF were shown as there is no data showing that heparin-coupled bFGF activates fibroblasts better and there is little support in the data that coupling in general has a better effect than bFGF in solution: fig 1d shows best effect for bFGF in solution and minor effects for covalently bound bFGF, whereas fig 1c shows a better effect for covalently bound bFGF at 100ng in hMSCs, but the authors don't use this condition later to test whether these cells show higher FGFR1 expression.
- 5th paragraph: microfibrillar substrates
- Please add the figure reference showing the FGFR1 expression levels of cells on ESP scaffolds compared to TCP
- Page 5, top: the claim about the role of FA formation on ESP scaffolds is a bit misleading, as hMSCs show already a clear change in FA when plated on films, i.e. the chemical properties of the substrate seem to be as or more important than the form (film or scaffold).
- Section 'FGFR1 is not regulated through zyxin or paxillin': please clarify also in the text under which culture conditions the depletion experiments were performed. Did the authors try to attach fibronectin to the ESPs to foster formation of FAs and perhaps induce FGFR1 expression?
- Please add a reference to support the statement: MRTF translocates to the nucleus when actin is incorporated into actin filaments and globular actin is low, where it activates SRF to transcribe target genes.
- Section 'Actin-myosin and MRTF/SRF pathway regulate FGFR1 expression': blebbistatin keeps myosin II in the rigor state, i.e. usually attached to actin filaments. It is not an inhibitor of actin myosin interaction.
- Section 'Conclusion': the second sentence does not make sense.

Reviewer #2 (Remarks to the Author):

The manuscript from the Moroni lab proposes an interesting and novel characterization of the mechanosensitive nature of the bFGF-FGFR1-MTF-SRF pathway.

I believe the paper is worthy of being published but I would suggest two areas that I believe should be improved upon beforehand.

1. Manuscript organization.

The interesting observations in mechanobiology are made possible by the initial finding that hMSC and dermal fibroblasts behave differently when exposed to bFGF tethered to nanofibers (or 2D substrates). I got the impression that the initial goal of the study was perhaps to use tethered bFGF for applications in tissue engineering and cancer biology. After the observation was made, the study pivoted on the mechanobiology of FGFR1. Unfortunately, the text reads like it's trying to cover both the old and the new storylines. As a consequence, the introduction and the first few result sections are very confusing (see below for specific suggestions).

I suggest clearly re-organizing the manuscript providing:

- An introduction that covers only what's known about the biology of FGFR1 and points out potential links to mechanosensitivity. The authors can then formulate the mechanosensitivity hypothesis explicitly and argue that a way to test the hypothesis is to tether bFGF to substrates that can be engineered to control the mechanical microenvironment.
- The first result section should then simply cover the fabrication of the various substrates and the characterization of the behavior of hMSC and fibroblasts when exposed to said substrates.
- The rest of the manuscript will then follow normally but with a few minor suggestions (see below)

2. The big question mark with this study is whether the different behaviors in hMSCs and fibroblasts are specifically and exclusively due to the identified mechanosensitive axis and not associated with, or modulated by, phenotypic differences between the two distinct cell populations. This is made worse by the fact that there are several different preparations being used throughout the study and their mechanical (e.g., stiffness) and biological (e.g., effective FGF coverage) properties are not always clearly presented. Normally, I would suggest genetic manipulation to demonstrate that the hMSCs can be made to grow as fibroblasts, or vice-versa. Given the tragic situation with COVID-19, these additional experiments would be impossible for the authors. Therefore, I recommend devoting a strong paragraph in the Discussion to address this issue, explicitly.

Specific issues:

#1. In the Abstract, the authors use the abbreviation ESP for electrospun before its introduction later in the discussion. I'd avoid the abbreviation in the introduction.

#2. In the introduction, the authors state that ESP scaffolds "are particularly interesting for defects of limited depth, but with a large surface area." This comment would need to be qualified with a reference but it also adds relatively little to the mechanosensitive story. In fact, this comment is a good example of what could be removed to clarify the introduction by focusing it only on mechanosensation and FGFR1.

#3. In the introduction, the authors state "Sufficient number of cells and the right cell density is highly important for tissue engineering applications, so cell proliferation is a key aspect to control. Several growth factors are well known for their proliferation inducing abilities. Arguably, the most well-studied of these is basic fibroblast growth factor (bFGF). bFGF is known to increase proliferation rates in a wide variety of cell types and has anti-apoptotic effects, while maintaining or enhancing differentiation- and regeneration potential [6]." This type of comment, instead, should be clarified and expanded upon - since the authors will go on to show that FGF in fact will not induce proliferation in hMSC when tethered to fibers .

#4. Introduction. The sentence "Very little is known about the regulation of any FGFR, while a better understanding could advance the understanding of tumor development and open up new therapeutic targets." is broken and needs rewriting.

#5. The last paragraph in the introduction is a great summary of the study "Here, we have found

that hMSCs don't respond to soluble or covalently bound bFGF when cultured on microfibrillar substrates such as those created by ESP, while fibroblasts do. hMSCs, but not fibroblasts, downregulate FGFR1 expression when cultured on ESP scaffolds. We show that FGFR1 expression is mechanosensitive and works through actin-myosin tension and the MRTF/SRF pathway. Inhibition of the MRTF/SRF pathway made hMSCs irresponsive to bFGF on tissue culture plastic (TCP) and downregulated FGFR1 in hMSCs, fibroblasts and MG63 osteosarcoma cells." Unfortunately, the text prior to it does little to establish a framework in which readers can appreciate this summary. See my general comment on a potential re-organization of the introduction.

#6. Result. The sentence "300PEOT45PBT55 was used to produce 50 μ m thick ESP scaffolds with 0.99 \pm 0.18 μ m average fiber diameter" starts with the alpha-numeric polymer name, which reads early. Consider rewriting.

#7. Results. "With the addition of EDC-NHS, around 98% of bFGF was bound to the scaffolds, aspecifically or covalently. Before cell culture, scaffolds were thoroughly washed in water and PBS, to wash away most of the a-specifically bound bFGF." The presence of potential a-specific binding is a concern, which is made worse as there are several preparations and several rounds of EDC-NHS a-specific chemistry and washing being performed across the various preparations *see also #9). I think a supplemental figure summarizing how much of FGF ends up on the various preparations right before cell seeding would be important to make sure comparisons are run in a fair manner.

#8. Results. "PEG-NH₂ was incorporated into the electrospinning polymer solution to introduce amino groups on the surface of the ESP scaffold." Does this additional step alter substrate stiffness appreciably? Again, a supplemental figure comparing substrate stiffness across preparations would be helpful to establish whether mechanical stiffness is a confounding factor.

#9. Results. "bFGF was then bound to the heparin-functionalized scaffolds by overnight incubation. As the heparin interfered with the bFGF ELISA (data not shown), the amount of absorbed bFGF could not be measured." See also #7.

#10. Results. "To test why fibroblasts did, but hMSCs did not, respond to bFGF when cultured on such microfibrillar substrates" here the substrates are called microfibrillar but earlier they were ESP scaffolds or nanofibers. I'd suggest a consistent terminology, unless, fibers need to be "micro" in this instance, which should be explained in more details.

#11. Results. "Aberrant FGFR regulation in cancer cells has been linked to metastasis, tumor progression and a worse diagnosis. To test whether the MRTF/SRF pathway is also responsible for FGFR1 regulation in cancer cells, we treated the osteosarcoma cell line MG63 with the MRTF/SRF inhibitor. Similar to hMSCs and fibroblasts, FGFR1 expression was reduced by 60.2 \pm 10.3% (p<0.05) when MRTF/SRF was inhibited (Fig. 5f). MRTF/SRF inhibition decreased FGFR1 expression in 3 different human cell types, suggesting that the MRTF/SRF pathway is a univocal regulator of the FGFR1 pathway." This example with cancer cells comes completely out of nowhere. It should either be developed into its own result sections, to support the potential clinical relevance of this study or moved to supplement.

#12. Discussion. "Here, we have functionalized 300PEOT45PBT55 ESP scaffolds by coupling bFGF to the surface. The covalent binding of bFGF to ESP scaffolds made of other polymers has been shown before to retain the growth factor bioactivity [10, 50]." After the summary, the discussion should address the fact that the material itself (300PEOT45PBT55) and is capable of limiting hMSC proliferation but the effect is potential in fibers.

#13. Discussion. "Similarly, the covalently coupled bFGF was still active on our ESP scaffolds, and

could be used as a method to increase cell proliferation rate on ESP scaffolds." This should mention fibroblast proliferation, not general cells, since there might be other cell types that behave like hMSCs.

#14. Discussion. "MRTF-A was located in the nucleus of fibroblasts on all substrates, and of hMSCs on TCP, strongly suggesting together with high SRF activity that the MRTF/SRF pathway was active. " The TCP abbreviation can be substituted with tissue culture plastics to facilitate reading.

Dear editor and reviewers,

We thank the reviewers for their detailed look to our manuscript and the valuable suggestions they have given. Due to the COVID-19 situation, we are unable to perform more experiments in the foreseeable future to further expand this manuscript. Therefore, we have discussed with the editor that a rigorous reorganization of the manuscript could also be sufficient, as suggested by reviewer #2. We have reorganized the manuscript to focus on the biological aspect of this work, rather than the tissue engineering, we have changed certain claims and figures and added a quantification of MRTF, which was still possible with the current data. We hope that together, these changes have improved the clarity and impact of our manuscript. Below we have responded in detail to each comment and all changes are marked with track changes in the manuscript document.

Referee expertise:

Referee #1: Biomechanics of immunology

Referee #2: Signalling in biomechanics and mechanobiology

Reviewers' comments:

Reviewer #1 (Remarks to the Author):

The article 'Mechanosensitive regulation of FGFR1 through the MRTF-SRF pathway' report about the lack of hMSCs to respond to bFGF signaling when cultured on ESP scaffolds and conclude with the help of further experiments and comparison to fibroblast cells that this lack of bFGF signaling is caused by a mechanosensitive downregulation of FGFR1 expression via the MRTF/SRF pathway.

The authors address an interesting topic that could be relevant for the development and improvement of techniques to artificially grow tissues and that would need more investigation. However, the presented results are not supporting the claims sufficiently to justify publication at the current stage and further experiments would be required. It would be helpful if the authors could structure the introduction and discussion better to clarify the motivation of the study (e.g. why using hMSCs in ESP scaffolds) and how the findings could help the wider community.

The methods are well described, and statistical analysis was performed to a sufficient level (see comments).

Major comments:

1. The study does not provide any direct evidence for the claim that FGFR1 regulation is mechanosensitive through the MRTF-SRF pathway. The authors show that FGFR1 expression is reduced in hMSCs on ESP scaffolds and that the MRTF-SRF pathway is downregulated in hMSCs on ESP scaffolds. But there is no evidence that this is mediated via a mechanosensitive pathway. There are not measures of the mechanical properties of ESP scaffolds or films and no measures indicating a different acto-myosin contractility in hMSCs on ESP scaffolds or films. If it would be a mechanosensitive pathway, then the alteration of

the mechanical properties of the environment should be sufficient to explain the difference in signaling, i.e. the authors would have to provide evidence, that the ESP films and scaffolds provide a mechanically different environment to hMSCs leading to changes in acto-myosin activity/ contractility, and hence changes in MRTF localization or SRF levels.

Otherwise put: if the regulation would solely be caused by the mechanical differences of ESP scaffolds, one would expect no or very little changes in FGFR1 expression between hMSCs on TCP and ESP films.

Reply: The reviewer is correct that direct evidence for mechanosensitive regulation of FGFR1 is not given in this manuscript. The claim has been changed throughout the manuscript as well as in the title.

2. To link acto-myosin activity with MRTF regulated FGFR1 expression, the authors treat cells with blebbistatin and observe 40-50% reduction of FGFR1 expression in both, fibroblasts and hMSCs on TCP cultures. However, in order to link this reduction to the MRTF pathway, the authors should report the localization of MRTF (nucleus vs cytosolic) upon blebbistatin treatment.

Reply: With the current data the blebbistatin and MRTF data are indeed not directly linked. Others have, however, shown that blebbistatin treatment leads to cytoplasmic localization of MRTF-A, in vitro^[1] and in vivo^[2]. Still, we can't definitively claim that both inhibitors act (solely) on the same pathway that inhibits FGFR1. Due to the current covid-19 situation, we can't perform these experiments, so we have avoided to make this claim in the manuscript.

It would also be good to compare the results with other inhibitors, e.g. LatA, that captures g-actin or jasplakinolide that stabilizes actin filaments.

Reply: This would be a nice addition to specifically go to the role of actin, rather than actin-myosin. However, as we're unable to perform more experiments, we kept the focus on actin-myosin.

3. The data on MRTF localization and SRF levels as function of the substrate hMSCs are cultured on, is not very conclusive. The authors state in the discussion, that hMSCs on ESP scaffolds display high nuclear localization of MRTF, but SRF levels show a clear trend which would counter their earlier argument that the FGFR1 expression is regulated via MRTF.

Reply: It is indeed possible that there is MRTF independent gene expression of SRF, although almost all SRF target genes are regulated by MRTF^[3]. We have adapted the discussion to include this point.

In addition, there is little discussion why the fibroblasts don't show a reduction in FGFR1 expression while displaying a reduction in SRF levels. It might help to quantify the relative localization and to move the images to the main figure.

Reply: The reduction of SRF in the fibroblasts is not significant. The trend is mentioned in the results section, but we don't think it would be good to discuss this, as statistically there is no difference.

We have quantified the MRTF localization in all conditions and have moved this to the main figures as suggested. Figures 4, 5 and 6 have also been reorganized to go along with this change.

4. Display of data and statistics: it would be clearer to use box plots (including data points)

instead of the chosen bar diagrams to display the spread of the measures. In connection to that, since the majority of the data is obtained from a small number of samples ($n = 3-5$) and the variation of bot read outs is high, it is advisable to state the relative changes in full numbers only (e.g. $87 \pm 6 \%$ instead of $86.5 \pm 5.6 \%$).

Reply: All graphs have been changed to include individual data points. Also, the relative changes are now displayed only as full numbers in the text.

Minor comments:

Introduction:

- 5th Paragraph: ...receptors (coming from 4 FGFR genes, FGFR1-4); tyrosine kinase receptors that can activate a variety... The phrase does not make sense. There seems to be something missing in between.

- Same paragraph: the last two phrases can be combined.

- 6th Paragraph, first phrase: development

Reply: We thank the reviewer for the detailed look. We have corrected the mentioned sections.

Results:

- 1st paragraph would fit better into the introduction

- 2nd paragraph: EDC-NHS chemistry: Please write out abbreviations when they are used first in the text

- ...A 26.4 ± 0.8 fold (p...) : it's clearer to use 'fold' instead of 'x', be consistent with the separators (it's altering between ', ' and '. '), write directly the measured number and not ~27 fold (...).

Reply: The mentioned sectioned are corrected.

- Better formulation: After washing with SDS to remove unbound BSA... (you might not remove unspecifically binding BSA with the wash)

Reply: We have used less strong wording now.

- 3rd paragraph: ... aspecifically to the scaffolds at all three concentrations

Reply: Corrected.

- 4th paragraph: can you give an estimation of the surface density of bFGF for the different concentrations at which bFGF was coupled to the ESP scaffolds?

Reply: Unfortunately, we can't. For this we would have to measure the total surface area for the electrospun scaffold, which is difficult on its own. And then measure the amount of bFGF that is covalently bound to the surface, which is also very difficult on a 3D microfibrillar scaffold such as ESP scaffolds. What we can say is that approximately 3, 30 and 300 ng bFGF has covalently coupled to the surface, assuming that the addition of EDC-NHS does not alter the aspecific binding of bFGF (figure 1b).

- Reference to figure is missing for the DNA increase between non-functional and bFGF coupled ESP scaffolds.

Reply: Corrected.

- It's not clear why the tests with the heparin-coupled bFGF were shown as there is no data showing that heparin-coupled bFGF activates fibroblasts better and there is little support in

the data that coupling in general has a better effect than bFGF in solution: fig 1d shows best effect for bFGF in solution and minor effects for covalently bound bFGF,

Reply: We started the study with trying to couple functional bFGF to the surface of ESP scaffolds. We show that the bFGF is still functional, as the fibroblasts respond to the covalently bound bFGF (fig 1d). The hMSCs did not respond to either soluble or bound bFGF. As heparin has been shown to help with the signaling of bFGF^[4], we thought it would be a nice additional experiment. We agree that this is not a major experiment for the paper, but we think it's a sufficiently nice addition to be included in the paper.

whereas fig 1c shows a better effect for covalently bound bFGF at 100ng in hMSCs, but the authors don't use this condition later to test whether these cells show higher FGFR1 expression.

Reply: Even though there is a trend, the increase in DNA of the covalently bound bFGF compared to the control conditions is not significant. Therefore, we did not think it warranted further investigation into the FGFR1 expression in that specific condition.

- 5th paragraph: microfibrillar substrates

- Please add the figure reference showing the FGFR1 expression levels of cells on ESP scaffolds compared to TCP

Reply: Corrected.

- Page 5, top: the claim about the role of FA formation on ESP scaffolds is a bit misleading, as hMSCs show already a clear change in FA when plated on films, i.e. the chemical properties of the substrate seem to be as or more important than the form (film or scaffold).

Reply: For the expression of zyxin, the films indeed seem to reduce zyxin and paxillin expression in hMSCs. However, when looking at the actual focal adhesion formation of both zyxin and paxillin, there is no difference between TCP and Films, but a clear difference of hMSCs on ESP scaffolds (Figure 3c,d and Supplementary figure 4c). We therefore think it is safe to state that the ESP scaffolds change focal adhesion formation. Indeed, the material itself might play a role in this, as the films also influence expression of both zyxin and paxillin, even though they still assemble into large focal adhesions on films.

- Section 'FGFR1 is not regulated through zyxin or paxillin': please clarify also in the text under which culture conditions the depletion experiments were performed.

Reply: As the hMSCs on ESP scaffolds already displayed little focal adhesions, we did these experiments on TCP, to create a contrast between many (scramble shRNA on TCP) and almost none (ZYG or PXN shRNA on TCP). This has been clarified in the text.

Did the authors try to attach fibronectin to the ESPs to foster formation of FAs and perhaps induce FGFR1 expression?

Reply: We did not perform this experiment because we proved that zyxin or paxillin do not influence FGFR1 expression. We therefore investigated another possible reason for the difference in FGFR1 expression on ESP scaffolds.

- Please add a reference to support the statement: MRTF translocates to the nucleus when actin is incorporated into actin filaments and globular actin is low, where it activates SRF to transcribe target genes.

- Section 'Actin-myosin and MRTF/SRF pathway regulate FGFR1 expression': blebbistatin keeps myosin II in the rigor state, i.e. usually attached to actin filaments. It is not an inhibitor of actin-myosin interaction.
- Section 'Conclusion': the second sentence does not make sense.

Reply: The mentioned sections have been corrected.

Reviewer #2 (Remarks to the Author):

The manuscript from the Moroni lab proposes an interesting and novel characterization of the mechanosensitive nature of the bFGF-FGFR1-MTF-SRF pathway.

I believe the paper is worthy of being published but I would suggest two areas that I believe should be improved upon beforehand.

1. Manuscript organization.

The interesting observations in mechanobiology are made possible by the initial finding that hMSC and dermal fibroblasts behave differently when exposed to bFGF tethered to nanofibers (or 2D substrates). I got the impression that the initial goal of the study was perhaps to use tethered bFGF for applications in tissue engineering and cancer biology. After the observation was made, the study pivoted on the mechanobiology of FGFR1.

Unfortunately, the text reads like it's trying to cover both the old and the new storylines. As a consequence, the introduction and the first few result sections are very confusing (see below for specific suggestions).

I suggest clearly re-organizing the manuscript providing:

- An introduction that covers only what's known about the biology of FGFR1 and points out potential links to mechanosensitivity. The authors can then formulate the mechanosensitivity hypothesis explicitly and argue that a way to test the hypothesis is to tether bFGF to substrated that can be engineered to control the mechanical microenvironment.

Reply: We thank the reviewer for the detailed look and the good suggestions on reorganizing the manuscript. Accordingly, we have greatly changed and reorganized the introduction, result sections and discussion to focus on bFGF response and FGFR1 regulation, rather than tissue engineering.

- The first result section should then simply cover the fabrication of the various substrates and the characterization of the behavior of hMSC and fibroblasts when exposed to said fibroblasts.

Reply: We have rewritten this section to focus on the biological questions, rather than the tissue engineering applications.

- The rest of the manuscripts will then follow normally but with a few minor suggestions (see below)

2. The big question mark with this study is whether the different behaviors in hMSCs and fibroblasts are specifically and exclusively due to the identified mechanosensitive axis and not associated with, or modulated by, phenotypic differences between the two distinct cell populations. This is made worse by the fact that there are several different preparations being

used throughout the study and their mechanical (e.g., stiffness) and biological (e.g., effective FGF coverage) properties are not always clearly presented. Normally, I would suggest genetic manipulation to demonstrate that the hMSCs can be made to grow as fibroblasts, or vice-versa. Given the tragic situation with COVID-19, these additional experiments would be impossible for the authors. Therefore, I recommend devoting a strong paragraph in the Discussion to address this issue, explicitly.

Reply: We have expanded the discussion to more elaborately discuss this point.

Specific issues:

#1. In the Abstract, the authors use the abbreviation ESP for electrospun before its introduction later in the discussion. I'd avoid the abbreviation in the introduction.

Reply: This has been adjusted

#2. In the introduction, the authors state that ESP scaffolds "are particularly interesting for defects of limited depth, but with a large surface area." This comment would need to be qualified with a reference but it also add relatively little to the mechanosensitive story. In fact, this comment is a good example of what could be removed to clarify the introduction by focusing it only on mechanosensation and FGFR1.

Reply: The quoted sentence has been removed.

#3. In the introduction, the authors state "Sufficient number of cells and the right cell density is highly important for tissue engineering applications, so cell proliferation is a key aspect to control. Several growth factors are well known for their proliferation inducing abilities. Arguably, the most well-studied of these is basic fibroblast growth factor (bFGF). bFGF is known to increase proliferation rates in a wide variety of cell types and has anti-apoptotic effects, while maintaining or enhancing differentiation- and regeneration potential [6]." This type of comments, instead, should be clarified and expanded upon - since the authors will go on to show that FGF in fact will not induce proliferation in hMSC when tethered to fibers .

Reply: This has been adjusted, along with a greater change in the introduction

#4. Introduction. The sentence "Very little is known about the regulation of any FGFR, while a better understand could advance the understanding of tumor development and open up new therapeutic targets." is broken and needs rewriting.

Reply: Corrected.

#5. The last paragraph in the introduction is a great summary of the study "Here, we have found that hMSCs don't respond to soluble or covalently bound bFGF when cultured on microfibrillar substrates such as those created by ESP, while fibroblasts do. hMSCs, but not fibroblasts, downregulate FGFR1 expression when cultured on ESP scaffolds. We show that FGFR1 expression is mechanosensitive and works through actin-myosin tension and the MRTF/SRF pathway. Inhibition of the MRTF/SRF pathway made hMSCs irresponsive to bFGF on tissue culture plastic (TCP) and downregulated FGFR1 in hMSCs, fibroblasts and MG63 osteosarcoma cells." Unfortunately, the text prior to it does little to establish a framework in which readers can appreciate this summary. See my general comment on a potential re-organization of the introduction.

Reply: The introduction has been restructured.

#6. Result. The sentence "300PEOT45PBT55 was used to produce 50µm thick ESP scaffolds

with $0.99 \pm 0.18 \mu\text{m}$ average fiber diameter" starts with the alpha-numeric polymer name, which reads early. Consider rewriting.

Reply: The sentence has been modified.

#7. Results. "With the addition of EDC-NHS, around 98% of bFGF was bound to the scaffolds, aspecifically or covalently. Before cell culture, scaffolds were thoroughly washed in water and PBS, to wash away most of the a-specifically bound bFGF." The presence of potential a-specific binding is a concern, which is made worse as there are several preparations and several rounds of EDC-NHS a-specific chemistry and washing being performed across the various preparations *see also #9). I think a supplemental figure summarizing how much of FGF ends up on the various preparations right before cell seeding would be important to make sure comparisons are run in a fair manner.

Reply: We were unable to measure the amount of bFGF on the scaffold after all washings. For this reason, we've used the ELISA approach and measured what was left in solution after the coupling. The controls where bFGF is added without EDC-NHS (aspecific absorption) are also taken along in the cell experiments. This condition does not increase cell proliferation by itself, showing that without covalent binding, not enough bFGF is absorbed and/or released to activate more proliferation. We only saw an increase in fibroblast proliferation in the highest concentration of covalently coupled bFGF. Even though aspecifically bound bFGF on its own had no effect, we cannot exclude that there is a combinatorial effect of aspecifically and covalently bound bFGF in the covalently bound bFGF condition. The first paragraph of the discussion has been expanded to include this point.

#8. Results. "PEG-NH₂ was incorporated into the electrospinning polymer solution to introduce amino groups on the surface of the ESP scaffold." Does this additional step alter substrate stiffness appreciably? Again, a supplemental figure comparing substrate stiffness across preparations would be helpful to establish whether mechanical stiffness is a confounding factor.

Reply: For the heparin experiments, all conditions were done with PEG-NH₂ scaffolds, also the non-functionalized scaffold. The addition of 1% PEG in the polymer solution might alter scaffold properties, but this should be equal for all conditions. A heparin only control was also taken along. Neither conditions improved proliferation of hMSCs. The text has been slightly modified to make this more clear.

#9. Results. "bFGF was then bound to the heparin-functionalized scaffolds by overnight incubation. As the heparin interfered with the bFGF ELISA (data not shown), the amount of absorbed bFGF could not be measured." See also #7.

Reply: See reply to #7.

#10. Results. "To test why fibroblasts did, but hMSCs did not, respond to bFGF when cultured on such microfibrillar substrates" here the substrates are called microfibrillar but earlier they were ESP scaffolds or nanofibers. I'd suggest a consistent terminology, unless, fibers need to be "micro" in this instance, which should be explained in more details.

Reply: The text has been made consistent and the scaffolds are now called microfibrillar substrates throughout the text.

#11. Results. "Aberrant FGFR regulation in cancer cells has been linked to metastasis, tumor progression and a worse diagnosis. To test whether the MRTF/SRF pathway is also

responsible for FGFR1 regulation in cancer cells, we treated the osteosarcoma cell line MG63 with the MRTF/SRF inhibitor. Similar to hMSCs and fibroblasts, FGFR1 expression was reduced by $60.2 \pm 10.3\%$ ($p < 0.05$) when MRTF/SRF was inhibited (Fig. 5f). MRTF/SRF inhibition decreased FGFR1 expression in 3 different human cell types, suggesting that the MRTF/SRF pathway is a univocal regulator of the FGFR1 pathway." This example with cancer cells comes completely out of nowhere. It should either be developed into its own result sections, to support the potential clinical relevance of this study or moved to supplement.

Reply: We have moved the MG63 data to the supplementary figures.

#12. Discussion. "Here, we have functionalized 300PEOT45PBT55 ESP scaffolds by coupling bFGF to the surface. The covalent binding of bFGF to ESP scaffolds made of other polymers has been shown before to retain the growth factor bioactivity [10, 50]." After the summary, the discussion should address the fact that the material itself (300PEOT45PBT55) and is capable of limiting hMSC proliferation but the effect is potential in fibers.

Reply: The end of the second paragraph of the discussion has been expanded to discuss this.

#13. Discussion. "Similarly, the covalently coupled bFGF was still active on our ESP scaffolds, and could be used as a method to increase cell proliferation rate on ESP scaffolds." This should mention fibroblast proliferation, not general cells, since there might be other cell types that behave like hMSCs.

Reply: "cell" has been changed to "fibroblast" and a section has been added to further elaborate on this.

#14. Discussion. "MRTF-A was located in the nucleus of fibroblasts on all substrates, and of hMSCs on TCP, strongly suggesting together with high SRF activity that the MRTF/SRF pathway was active. " The TCP abbreviation can be substituted with tissue culture plastics to facilitate reading.

Reply: We could change all mention of TCP to tissue culture plastic, although we would prefer to write it as TCP, as this abbreviation is commonly used in literature. We let the editor decide on this point.

References

1. L.S. Hinojosa, M. Holst, C. Baarlink, and R. Grosse, *MRTF transcription and Ezrin-dependent plasma membrane blebbing are required for entotic invasion*. Journal of Cell Biology, 2017. **216**(10): p. 3087-3095.
2. N. Gjorevski, A.S. Piotrowski, V.D. Varner, and C.M. Nelson, *Dynamic tensile forces drive collective cell migration through three-dimensional extracellular matrices*. Scientific reports, 2015. **5**: p. 11458-11458.
3. C. Esnault, A. Stewart, F. Gualdrini, P. East, S. Horswell, N. Matthews, and R. Treisman, *Rho-actin signaling to the MRTF coactivators dominates the immediate transcriptional response to serum in fibroblasts*. Genes & development, 2014. **28**(9): p. 943-958.
4. M.A. Caldwell, E. Garcion, M.G. terBorg, X. He, and C.N. Svendsen, *Heparin stabilizes FGF-2 and modulates striatal precursor cell behavior in response to EGF*. Exp Neurol, 2004. **188**(2): p. 408-20.

Reviewers' comments:

Reviewer #1 (Remarks to the Author):

Given the current Covid-19 situation it is totally understandable that no additional experiments were possible, but the authors have done a great job in restructuring the article and improving the clarity of the manuscript. All points that were raised by the reviewers were addressed satisfactorily and I could not find any parts that would need correction or clarification. I would recommend the article for publication.

Reviewer #2 (Remarks to the Author):

The revised version of this manuscript from the Moroni's group addressed most of my criticism. I am particularly impressed with the molecular biology in this study but fabrication and characterization details, as well as the overall clarity and writing of the manuscript could be improved.

I am attaching a revised word document (the name of the reviewer was removed) to indicate a number of possible useful changes.

My two main criticisms are as follows:

1. The connection with cancer is really confusing. Entire paragraphs in the Intro and Discussion are devoted to cancer but, at present, there are only two panels in a supplemental figure worth of results obtained from a single cell line. If the connection with cancer is important, more data is needed. If not, the storyline can be removed entirely and the introduction and discussion can be dedicated only to the regenerative medicine storyline. Importantly, regenerative medicine is a stronger application of this research since fibrous scaffolds are often employed (alone or in combination with cells) as regenerative interventions.

2. I am still confused by the quantification of bFGF tethering via ELISA (text and figure 1). As I indicated in the word document, it might be the writing. If I understand correctly, three known concentrations of bFGF were tested with and without EDC-NHS chemistry. The washout solution is then tested with ELISA to measure the bFGF that bounded α -specifically and, therefore, was released. A few considerations:

a. is there any bFGF left in the bathing solution before the washing steps? Was this tested with ELISA? If the largest known concentration in the bathing solution is 1000 ng but only 300 ng binds to the fibers, the left-over from ELISA should be normalized against this real value. I don't think this is the case (see point b) but it's unclear from the text.

b. that the largest of the concentrations used (1000 ng) has essentially the same amount of unbound bFGF after washout (~ 0.1 ng from the graphs) then the smallest concentration (10 ng). Given the long incubation time (overnight) after EDC-NHS, I would expect most bFGF to bind which suggests 1000 ng is not sufficient to saturate the fibers. Or, alternatively (see point a) that not all 1000 ng bind before washout.

c. since bFGF in solution was used at a concentration of 10 ng/mL (but we don't know the total volume) it's unclear how the tethered concentrations compare to the fiber deposited ones.

The big problem here is that if bFGF is not saturating the fibers, an unknown percentage of the cells in that experimental group might not be exposed to bFGF which will invalidate the

comparison. For example, if only 30% of the fiber surface area is coated but the DNA is collected from confluent cells, ~70% of the collected DNA will come from cells that were not exposed to bFGF.

Ultimately, I believe the experiments were done correctly but it is not clear from the current version of the manuscript. The authors should clarify the methodologies and the various steps used in this set of experiments. If additional ELISA experiments are still not possible, I would suggest a simple theoretical model of bFGF binding that can be used to estimate how much bFGF is needed to saturate the surface of the fibers. With such model, the experimental concentrations of bFGF used in the manuscript can be placed in context.

A number of minor comments and edits are in the attached revised document.

Regulation of FGFR1 through actin-myosin and the MRTF-SRF pathway

Jip Zonderland, Silvia Rezzola, Lorenzo Moroni

Complex Tissue Regeneration Department, MERLN Institute for Technology-Inspired
Regenerative Medicine, Maastricht University, Maastricht, the Netherlands.

Abstract

Controlling basic fibroblast growth factor (bFGF) signaling is important for cell proliferation and differentiation in both regenerative medicine purposes, as well as controlling proliferation and differentiation potential, and for tumor progression and metastasis in cancer biology, influencing tumor progression and metastasis. Here, we observed that human mesenchymal stromal cells (hMSCs), but not fibroblasts, ~~no~~ lose the ability to respond longer responded to soluble or covalently bound bFGF when cultured on microfibrillar substrates, while fibroblasts did. This behavior correlated with a downregulation of FGF receptor 1 (FGFR1) expression of hMSCs on microfibrillar substrates, compared to hMSCs on conventional tissue culture plastic (TCP). hMSCs also expressed less SRF on microfibrillar substrates, compared to TCP, while fibroblasts maintained high FGFR1 and SRF expression. Inhibition of actin-myosin tension or the MRTF/SRF pathway decreased FGFR1 expression in hMSCs, fibroblasts and MG63 osteosarcoma cells. This downregulation was functional, as hMSCs became irresponsive to bFGF in the presence of MRTF/SRF inhibitor. Together, our data show that hMSCs, but not fibroblasts, are irresponsive to bFGF when cultured on microfibrillar substrates by downregulation of FGFR1 through the MRTF/SRF pathway. This is the first time FGFR1 expression has been shown to be regulated through a mechanosensitive pathway and adds to the sparse literature on FGFR1 regulation. These results could open up new targets for cancer treatments and could aid designing tissue engineering constructs that better control cell proliferation.

Introduction

Understanding proliferation is critical for regenerative medicine approaches and cancer biology. For regenerative medicine approaches, it is important to understand and control proliferation both *in vitro*, to obtain a sufficient number of cells for proliferation, and *in vivo*,

Comment [A1]: Are hMSC intended to be a model for regenerative medicine and fibroblasts a model for cancer biology? If not, this into paragraph is confusing.

I would rephrase with something like: "bFGF signaling modulates a variety of basic cellular functions, including proliferation and differentiation, which are important in regenerative medicine and cancer biology. Interestingly, the same cellular processes can be modulated by substrate mechanics and topography, so we asked whether there was a mechanosensitive component to bFGF-mediated cellular proliferation. We found that hMSCs, but not dermal fibroblasts, lost sensitivity to soluble or tethered bFGF when cultured on microfibrillar substrate. [...]"

Comment [A2]: Spell out.

Comment [A3]: Is this data still in the main manuscript? If it has been moved to supplement, I doubt the cell line should be mentioned in the abstract.

Comment [A4]: You absolutely sure?

to control cell growth and regeneration. Several growth factors are well known for their proliferation inducing abilities. Arguably, the most well-studied of these is basic fibroblast growth factor (bFGF). bFGF is known to increase proliferation rates in a wide variety of cell types and has anti-apoptotic effects, while maintaining or enhancing differentiation- and regeneration potential^[1]. bFGF can bind to 7 FGF receptors (FGFR) (coming from 4 FGFR genes, FGFR1-4). All FGFRs are tyrosine kinase receptors that can activate a variety of pathways, including the RAS-MAPK, PI3K-Akt, PLC γ and STAT pathways^[2]. Using next generation sequencing to analyze 4853 tumors, Helsten et al. found aberrations in FGFR in 7.1% of all tumors^[3]. In addition, increased expression of FGFR has been correlated with a bad prognosis, increased metastasis and tumor progression in a large variety of cancers^[4-8]. Indeed, animal studies and clinical trials are currently ongoing to test the effects of FGFR inhibitors on cancer treatment, showing promising initial results^[9-14]. Unraveling FGFR regulation could advance the understanding of tumor development and open up new therapeutic targets.

Besides the role in cancer development, FGFR are also important for regenerative medicine purposes. FGFR1 and 2 have been shown to be involved in adipo- and osteogenic differentiation in hMSCs^[15, 16]. FGFR3 is highly expressed in chondrocytes and involved in chondrogenesis^[17]. Only FGFR1 has been shown to be involved in hMSCs proliferation^[18], while the other receptors remain unstudied in this regard. For this reason, here we focused on the regulation of FGFR1 expression.

Very little is known about the regulation of any FGFR. YAP knockdown has been shown to decrease FGFR1 expression in lung cancer cells^[19] and neurospheres^[20]. Also, integrin α 6 has been shown to regulate FGFR1^[20]. Both YAP and integrins play an important role in mechanosensing^[21, 22], hinting at a potential mechanosensitive regulation of FGFR1. Cells adhere to their surrounding matrix or culture substrate through integrins^[23]. When enough force ~~can be is~~ applied, integrin clusters can bind to the actin cytoskeleton through large protein complexes called focal adhesions^[24], ~~therefore connecting directly~~ ~~On the other end, actin filaments can be attached~~ to other focal adhesions, or to the nucleus^[25]. Between these attachment points, force can be generated by actin-myosin ~~filaments interactions that to~~ generate cellular tension^[26]. A large variety of cellular processes are regulated by cellular tension, including proliferation^[21, 27-29], differentiation^[30-32] and migration^[33]. Different transcription factors have been shown to orchestrate these changes in behaviors, of which

Comment [A5]: You cover a lot of ground in the intro and things don't follow smoothly. I suggest a bit more text here to introduce the various sub-topics/

"bFGF is a typical signal modulating cell proliferation via a family of FGF receptors, whose effect on various cell types including fibroblasts and hMSCs has been extensively studied in-vitro. At the same time, physical signals from the uEnvironment have also been shown to regulate cell proliferation in the same cell types via mechanosensitive signaling. Here, we investigate the cross-talk between these two pathways."

Comment [A6]: There keep being mentions of tumor and cancer biology → This sets up an expectation in the reader which will be frustrated when no cancer-related results are shown.

serum response factor (SRF) is a well-studied example. When globular actin concentrations are low in the cytoplasm, myocardin related transcription factor (MRTF) A or B enters the nucleus and binds to SRF to start transcribing target genes^[34].

As transplanted cells for regenerative medicine inevitably end up in a 3D environment, we wanted to investigate the potential mechanosensitive regulation of FGFR1 and response to bFGF in 3D. Previously, we have shown that hMSCs reduce cellular tension in 3D microfibrillar substrates and other 3D environments^[35]. Thus, to investigate the effect of bFGF in 3D and to potentially find leads on FGFR1 regulation, we started by investigating the response of hMSCs to bFGF in a 3D microfibrillar environment. In solution, bFGF, like most growth factors, is highly unstable and loses activity after 24-48 hours^[1, 36]. Covalently coupling bFGF to scaffolds has been shown to enhance stability while maintaining signaling activity^[37-39]. Therefore, we tested the response to bFGF in 3D for both soluble bFGF and covalently bound bFGF on microfibrillar substrates

Here, we ~~have~~ found that hMSCs ~~don't do not~~ respond to soluble or covalently bound bFGF when cultured on microfibrillar substrates ~~such as those created by ESP~~, while fibroblasts do. hMSCs, but not fibroblasts, downregulate FGFR1 expression when cultured on microfibrillar substrates. We show that FGFR1 expression is regulated through the mechanosensitive proteins actin-myosin and MRTF/SRF. ~~Further, the i~~nhibition of the MRTF/SRF pathway made hMSCs irresponsive to bFGF on tissue culture plastic (TCP) and downregulated FGFR1 in hMSCs, fibroblasts and MG63 osteosarcoma cells.

Results

Fibroblasts, but not hMSCs, respond to bFGF functionalized microfibrillar substrates

hMSCs were exposed to bFGF on microfibrillar substrates to study the effect of bFGF in a 3D environment. ~~To enhance stability of bFGF, we started by covalently coupling bFGF to the microfibrillar substrates^[37-39] and comparing to bFGF in the medium.~~

Microfibrillar substrates with a thickness of 50 μm and $0.99 \pm 0.18 \mu\text{m}$ average fiber diameter were produced by electrospinning 300PEOT45PBT55 (Supplementary Fig. 1). ~~To test the stability of bFGF in solution and tethered to the fibers, we need to covalently couple bFGF to the microfibrillar substrates^[37-39].~~ The ester bond in the polymer was opened using 0.5M NaOH to expose carboxyl groups on the surface of the scaffold. 1-ethyl-(dimethylaminopropyl)-carbodiimide (EDC) - N-hydroxysuccinimide (NHS) chemistry was

Comment [A7]: The potential link with Reg Med supports uFibrillar but not cancer. It is also very interesting and could be used earlier to motivate the study as a cross-talk between classical bFGF and mechanosensing

Comment [A8]: Again, if this cells appear only in one panel the all link with cancer is very weak

Comment [A9]: This should be moved after the description of the substrate

Comment [A10]: This isn't yet a comparison between soluble and tethered bFGF, just the introduction to the tethering method.

used to covalently couple the free amine groups of proteins to the surface of the scaffold.

~~As To validate our approach, we first use a model protein,~~ FITC ~~as a model protein to~~ labeled bovine serum albumin (BSA) ~~and then coupled FITC-BSA to was coupled to the~~ microfibrillar substrates. A 27 ± 1 fold ($p < 0.001$) higher fluorescent signal was observed when BSA was added after EDC-NHS, than when BSA was added after water control (Fig. 1a). After washing with SDS, to potentially wash away non-covalently bound BSA, the fluorescent signal was 40 ± 15 fold higher ($p < 0.001$) in the EDC-NHS group compared to BSA only. Together, this strongly suggests that covalent coupling of BSA was achieved.

Next, ~~we used this validated strategy to couple~~ bFGF ~~was coupled~~ to microfibrillar substrates, ~~using the same strategy~~. As opposed to bFGF in solution, cell response to covalently coupled bFGF has not been widely studied. In an attempt to find the right concentration range, we coupled three different amounts of bFGF to microfibrillar substrates. bFGF left over in solution after coupling was measured by ELISA (Fig. 1b).

Without the addition of EDC-NHS, around 70% of the bFGF adhered aspecifically ~~and could~~ ~~therefore be measured with the ELISA after the washing steps~~ to the scaffolds at all three concentrations. ~~With the addition of EDC-NHS, around 98% of bFGF was bound to the scaffolds, aspecifically or covalently. Before cell culture, scaffolds were thoroughly washed in water and PBS, to attempt to wash away aspecifically bound bFGF.~~

To test whether the bound bFGF was still functional, proliferation of hMSCs cultured on the microfibrillar substrates was assessed after 7 days (Fig. 1c). Interestingly, the hMSCs did not respond to either bFGF bound to the microfibrillar substrates, or bFGF in solution. In 2D tissue culture plastic, hMSCs did increase proliferation over 7 days in response to bFGF in solution, displaying $45 \pm 11\%$ more DNA, demonstrating that the microfibrillar environment influenced the hMSC's response to bFGF (Supplementary Fig. 2).

Fibroblasts are particularly well studied for their increase in proliferation in response to bFGF. To test whether this lack of response to bFGF when cultured on microfibrillar substrates was specific to hMSCs, human dermal fibroblasts were cultured for 7 days on the microfibrillar substrates. On non-functionalized scaffolds, $77 \pm 20\%$ more ($p < 0.0001$) DNA was found after 7 days of culture in the presence of bFGF in the medium (Fig. 1d). On the 1000 ng covalently coupled bFGF scaffolds, $50 \pm 13\%$ more ($p < 0.01$) DNA was found compared to non-functionalized scaffolds, showing that the covalently bound bFGF was still functional.

Comment [A11]: If I understand correctly from the response to reviewer

Comment [A12]: This approach keeps confusing me. It's worth discussing it earlier in the paragraph.

Try something like:
"To determine the extent of covalent binding to the fiber, we treat them with the three concentration of bFGF in the presence/absence of EDC/NHS. After binding, fibers were washed vigorously, and the amount of bFGF left-over post-washing was quantified via ELISA."

Comment [A13]: The way this is written begs for you to do ELISA on the washing media after the fact to measure bFGF.

That will allow you to estimate the portion of FGF that had bound a-specifically.

Alternatively, you could explicitly speculate on the final concentration of bFGF based on a simple model of saturation of the fiber surface (with a long incubation time and the EDC-NHS chemistry the simple model is probably pretty accurate)

Heparin is known to bind and stabilize bFGF and increase efficacy^[40]. To covalently couple heparin to the microfibrillar substrates, PEG-NH₂ was incorporated into the electrospinning polymer solution to introduce amino groups on the surface of the microfibrillar substrates.

~~The carboxyl groups of heparin were then bound to the microfibrillar substrates by EDC-NHS chemistry (Supplementary Fig. 3a). bFGF was then bound to the heparin functionalized scaffolds by overnight incubation. As the heparin interfered with the bFGF ELISA (data not shown), the amount of absorbed bFGF could not be measured. After 7 days of culture, no differences were observed between hMSCs cultured on heparin+bFGF scaffolds and the heparin only or non-functionalized PEG-NH₂ scaffolds (Supplementary Fig. 3b). This further demonstrates that hMSCs don't respond to bFGF on microfibrillar substrates, also not when bFGF is bound to heparin.~~

Together, these results show that ~~the covalently coupled bFGF to microfibrillar substrates was still functional~~ sustains proliferations in fibroblast, and but not in that hMSCs ~~do not respond to bFGF when cultured on microfibrillar substrates ESP scaffolds, but fibroblasts do.~~

Reduced FGFR1 expression on microfibrillar substrates in hMSCs, but not fibroblasts

To test why fibroblasts did, but hMSCs did not, ~~respond to bFGF when cultured on such microfibrillar substrates, we analyzed FGFR1 expression of hMSCs and fibroblasts cultured on TCP, as well as 2D films and microfibrillar substrates, and on 2D films made up of the same material as the microfibrillar substrates ESP.~~ Interestingly, when cultured on microfibrillar substrates, hMSCs expressed 87±5% less (p<0.01) FGFR1 than when cultured on TCP (Fig. 2a). On films, hMSCs displayed 67±7% less (p<0.01) FGFR1 expression than on TCP, showing that part of the reduction of FGFR1 expression on microfibrillar substrates comes from the material properties. However, on microfibrillar substrates the FGFR1 expression was still 60±16% lower (p<0.05) than on films, showing that regardless of material properties, the microfibrillar environment influenced FGFR1 expression.

Fibroblasts, however, did not display a difference in FGFR1 expression between the different culture substrates (Fig. 2b). The reduced FGFR1 expression of hMSCs on microfibrillar substrates, and the high FGFR1 expression of fibroblasts on microfibrillar

Comment [A14]: I suggest to replace this paragraph with a simple "[...] microfibrillar substrates but no significant differences were detected in the presence of Heparin (see suppl. Figure 3).

Comment [A15]: This phrasing is a bit abused throughout the text. It reads well in intro and discussion but I would change it in the results section with something like

"To test why fibroblasts and hMSCs responding differently to bFGF tethered to microfibrillar substrates, we...
"

substrates, potentially explains the difference in bFGF response of hMSCs and fibroblasts on microfibrillar substrates.

hMSCs, but not fibroblasts, display fewer focal adhesions on microfibrillar substrates

To understand why hMSCs, but not fibroblasts, reduced FGFR1 expression on microfibrillar substrates, we investigated ~~the focal adhesion distribution in difference in adhesion to the different substrates in~~ hMSCs and fibroblasts ~~by looking at focal adhesions~~. The expression of zyxin, an important focal adhesion protein, was reduced in both hMSCs and fibroblasts, respectively by $66\pm 7\%$ ($p<0.01$) and $79\pm 11\%$ ($p<0.05$) compared to TCP (Fig. 3a, b). Paxillin expression, another well studied focal adhesion protein, was significantly reduced in both hMSCs and fibroblasts on microfibrillar substrates, compared to TCP; respectively $73\pm 5\%$ ($p<0.01$) and $65\pm 8\%$ ($p<0.05$) (Supplementary Fig. 4a, b). On films, hMSCs also displayed reduced zyxin and paxillin expression, respectively $63\pm 17\%$ and $41\pm 11\%$ compared to TCP. Fibroblasts did not show a significant difference in zyxin or paxillin expression on films, compared to TCP.

When looking at the formation of zyxin positive focal adhesions, a reduction of $46\pm 18\%$ ($p<0.01$) of focal adhesions per cell area was observed when hMSCs were cultured on microfibrillar substrates, compared to TCP (Fig. 3c, d). When compared to films, hMSCs on microfibrillar substrates displayed $54\pm 16\%$ ($p<0.0001$) less zyxin positive focal adhesions per cell area. Interestingly, no significant difference was found between fibroblasts cultured on the different substrates (Fig. 3c, e). Indeed, when compared to fibroblasts grown on microfibrillar substrates, hMSCs on microfibrillar substrates displayed $60\pm 14\%$ ($p<0.0001$) fewer focal adhesions per cell area. The same trend was observed for paxillin positive focal adhesions, where hMSCs displayed far fewer paxillin positive focal adhesions on microfibrillar substrates than on films or TCP, while fibroblasts contained many paxillin positive focal adhesions on all three substrates (Supplementary Fig. 4c, d).

These results demonstrate that the microfibrillar environment changes focal adhesion formation in hMSCs, but not in fibroblasts. This shows that hMSCs adhere differently to the microfibrillar substrates than fibroblasts, potentially explaining the difference in FGFR1 expression.

Comment [A16]: Overused construct – see above.

As the lower FGFR1 expression correlated with fewer focal adhesions of hMSCs on microfibrillar substrates, we knocked down paxillin and zyxin in hMSCs cultured on TCP. Interestingly, neither paxillin nor zyxin depletion resulted in a change in FGFR1 expression, demonstrating that the differential expression of these proteins by hMSCs on microfibrillar substrates is not the reason for the reduced FGFR1 expression (Figure 4a-b).

SRF and MRTF-A correlate with FGFR1 expression

Even though focal adhesions ~~didn't~~ did not influence the FGFR1 expression, the reduction in focal adhesions of hMSCs on microfibrillar substrates suggests a difference in mechanosensitive signaling. An important mechanosensitive pathway is the MRTF/SRF pathway. MRTF translocates to the nucleus when actin is incorporated into actin filaments and globular actin is low, where it activates SRF to transcribe target genes^[34]. To investigate this pathway, we looked at the expression of SRF. Indeed, compared to TCP, the SRF expression was $73\pm 11\%$ ($p<0.05$) lower on films and $92\pm 6\%$ ($p<0.01$) lower on microfibrillar substrates in hMSCs. Compared to films, SRF expression was $70\pm 22\%$ ($p<0.05$) lower on microfibrillar substrates (Fig. 5a). For fibroblasts, expression of SRF was $25\pm 8\%$ ($p>0.05$) and $50\pm 20\%$ ($p>0.05$) lower on films and microfibrillar substrates respectively, but this difference was not statistically significant (Fig. 5b).

To further investigate the MRTF/SRF pathway, we looked at the localization of MRTF-A in hMSCs and fibroblasts, on TCP, films and microfibrillar substrates. In hMSCs on TCP, MRTF-A was located almost exclusively in the nucleus (Fig. 5c). On films, MRTF-A was found in the cytoplasm, with nuclear to cytoplasmic ratio of MRTF-A $93\pm 2\%$ less ($p<0.0001$) than on TCP (Fig. 5d). On microfibrillar substrates, MRTF-A was located in the nucleus and in the cytoplasm, with $45\pm 13\%$ less ($p<0.05$) nuclear translocation than on TCP. Similar to hMSCs, MRTF-A was located in the nucleus in fibroblasts on TCP. (Fig. 5e). On films and microfibrillar surfaces, MRTF-A was located both in the cytoplasm and in the nucleus, with $44\pm 14\%$ ($p<0.001$) and $47\pm 14\%$ ($p<0.001$) less nuclear translocation than on TCP, respectively (Fig. 5f).

Together with the SRF expression, these results suggest activity of the MRTF/SRF pathway in fibroblasts on all substrates and of hMSCs on TCP, but not of hMSCs on films or microfibrillar substrates. The activity of the MRTF/SRF pathway correlates with the reduced FGFR1 expression of hMSCs on films or microfibrillar substrates.

Actin-myosin and MRTF/SRF pathway regulate FGFR1 expression

To investigate the role of the MRTF/SRF pathway in the regulation of FGFR1 in hMSCs, we inhibited the pathway using CCG203971^[41, 42]. Indeed, in both hMSCs and fibroblasts, inhibition of the MRTF/SRF pathway reduced FGFR1 expression by $60\pm 7\%$ ($p < 0.01$) and $62\pm 3\%$ ($p < 0.01$), respectively (Fig. 6a, c). This shows that MRTF/SRF directly or indirectly regulates FGFR1 expression in both hMSCs and fibroblasts. We observed a strong decrease in SRF expression in hMSCs on microfibrillar substrates (Fig. 5a), strongly suggesting that the reduced FGFR1 expression of hMSCs on microfibrillar substrates is due to a decrease in SRF expression. Fibroblasts maintained a high expression of SRF on microfibrillar substrates (Fig. 5b), explaining which supports the high expression of FGFR1 on microfibrillar substrates.

When most actin monomers are assembled into filaments and globular actin is low, the MRTF/SRF pathway is activated. To determine the role of actin-myosin in the regulation of FGFR1, we treated hMSCs and fibroblasts with blebbistatin, a myosin inhibitor that greatly disrupts F-actin fibers. Expression of FGFR1 reduced $48\pm 10\%$ ($p < 0.05$) in hMSCs and $42\pm 13\%$ in fibroblasts (Fig. 6b, d). Together, this demonstrates that FGFR1 is regulated by the actin cytoskeleton and directly or indirectly through the MRTF/SRF pathway.

Another important mechanosensitive co-transcription factor is Yes activated protein 1 (YAP), entering the nucleus when a cell experiences high cellular tension^[21]. To investigate if YAP plays a role in FGFR1 regulation, we knocked down YAP in hMSCs. No difference was observed in FGFR1 expression between YAP-knock down and control-shRNA groups (Supplementary Fig. 5), demonstrating that YAP does not play a role in FGFR1 regulation in hMSCs.

To further investigate the link between the MRTF/SRF pathway and the FGF pathway, we investigated the response to bFGF of hMSCs cultured with MRTF/SRF inhibitor. After 7 days of culture on TCP in the presence of bFGF and/or MRTF/SRF inhibitor, total DNA was analyzed. As expected, $36\pm 9\%$ more DNA was found when bFGF was added to the medium, compared to basic medium (Fig. 6e). In the presence of MRTF/SRF inhibitor, $53\pm 5\%$ less DNA was found than in basic medium. Interestingly, in the presence of MRTF/SRF inhibitor, hMSCs did not increase proliferation when bFGF was added. This shows that the MRTF/SRF pathway regulates the response to bFGF, in confirmation with the reduced FGFR1 expression.

Comment [A17]: This bags for you to tell me that you did experiments to show whether it was directly or indirectly regulating actin...

Aberrant FGFR regulation in cancer cells has been linked to metastasis, tumor progression and a worse diagnosis. To test whether the MRTF/SRF pathway is also responsible for FGFR1 regulation in cancer cells, we treated the osteosarcoma cell line MG63 with the MRTF/SRF inhibitor. Similar to hMSCs and fibroblasts, FGFR1 expression was reduced by $60 \pm 10\%$ ($p < 0.05$) when MRTF/SRF was inhibited (Supplementary Fig. 6). MRTF/SRF inhibition decreased FGFR1 expression in 3 different human cell types, suggesting that the MRTF/SRF pathway is a univocal regulator of the FGFR1 pathway.

Discussion

Here, we ~~have~~ functionalized ESP 300PEOT45PBT55 microfibrillar substrates by coupling bFGF to the surface. The covalent binding of bFGF to the microfibrillar substrates made of other polymers has been shown before to retain the growth factor bioactivity^[39, 43]. Similarly, the covalently coupled bFGF was still active on our microfibrillar substrates and could be used as a method to increase fibroblast proliferation on microfibrillar substrates.

~~Other cells could also increase proliferation on these functionalized microfibrillar substrates, but it remains to be tested whether other cells respond like fibroblasts, or do not respond like hMSCs~~ This observation suggests that other cell types are likely to fall somewhere in the spectrum between highly- and non-proliferative when cultured on engineered substrates that recapitulate the fibrous micro environment of native tissues. This could be useful for *in vivo* approaches, but it can also be used as a cell culture substrate *in vitro*. bFGF is highly unstable in solution and covalent binding to a surface has been shown to increase its stability^[39]. Further investigation and characterization of the functionalized scaffolds would be useful for such uses. For example, the precise amount of covalently bound bFGF, the amount and effect of the potentially left over absorbed bFGF, and the stability of the covalently bound bFGF would be interesting parameters at the material interface to study.

Unlike fibroblasts, hMSCs did not increase proliferation in response to bFGF (in solution or covalently bound) on microfibrillar substrates. We found that this was due to reduced SRF expression, which caused decreased FGFR1 expression. SRF expression is known to be regulated by itself through a positive feedback loop^[44]. The observed difference in SRF expression between TCP, films and microfibrillar substrates highlight the difference in SRF activity on the different substrates. The positive feedback loop can exaggerate the differences in SRF expression, but the origin of the initial difference in SRF expression

Comment [A18]: I remain baffled by this cancer connection. In my opinion, either cancer is

#1 important to the storyline and it should have a dedicated figure (not a panel) and a place in the Intro and Discussion

#2 Not important for the story line, in which case it could be entirely removed

My personal preference is for option #2. The study is already interesting, has a strong focus on basic science, and a well-supported link to applications in regenerative medicine (via fibrous substrates increasingly used for prostheses). A passing reference to cancer with a single panel worth of data just distracts from the main storyline.

Comment [A19]: This needs to be checked for consistency – either say microfibrillar every time or ESP.

(Tip: Use ctrl+F to find all instances and fix)

Comment [A20]: I think you should consider explicitly setting up a model for this paper based on saturating the fiber surface with covalently bound bFGF as supported by trace amounts of bFGF found via ELISA after wash-out (as suggested in my earlier comment).

In the discussion, you could then suggest that the model could be refined in a further study and used to optimize manufacturing by reducing the amount of bFGF used. The optimization process would be important if you ever wanted to scale the technique for actual application in regenerative medicine

remains unclear. The 300PEOT45PBT55 material itself also affected FGFR1 and SRF expression, as seen by reduced FGFR1 and SRF expression in hMSCs on films vs TCP. The microfibrillar substrates further decreased this expression, on the same material, showing a direct effect of the microfibrillar environment on FGFR1 and SRF expression. MRTF-A was located in the nucleus of hMSCs and fibroblasts on TCP, suggesting, together with high SRF activity, that the MRTF/SRF pathway was active. Fibroblasts on films and microfibrillar substrates also displayed nuclear MRTF-A, although less than on TCP. As SRF expression didn't significantly change on the different substrates, this could explain the high FGFR1 expression and responsiveness to bFGF on microfibrillar substrates. hMSCs on films did not show nuclear localization of MRTF-A, which together with the low SRF expression suggests that the pathway is inactive, explaining the low FGFR1 expression. On microfibrillar surfaces, MRTF-A was located partly in the nucleus, although less than on TCP. Even though MRTF-A was located in the nuclei, the low SRF expression of hMSCs on microfibrillar substrates could prevent active transcription of the FGFR1 gene, or of genes that (indirectly) regulate FGFR1. We cannot exclude the possibility that MRTF-A does not play a role in FGFR1 regulation, although almost all SRF target genes are regulated by MRTF^[45]. As with FGFR1 and SRF expression, MRTF-A localization was also affected by the 300PEOT45PBT55 material as well as directly by the microfibrillar environment. Yet, it remains to be investigated through by which mechanism properties of the material in the 2D and microfibrillar form the material itself affected the expression of these proteins, and if it is through the same or a different mechanism than through which the microfibrillar substrates affect the expression, remains to be investigated.

Through actin-myosin inhibition by blebbistatin, we found that FGFR1 expression is reduced with less actin-myosin tension. The MRTF/SRF pathway is dependent on the actin cytoskeleton, but also plays a role in shaping the actin network^[46]. Whether the effect of actin-myosin inhibition went through MRTF/SRF, or vice-versa, we did not investigate. It is possible that no clear cause and effect between these two players exists, because there is a positive feedback loop between the two. MRTF/SRF activity increases stress fiber formation, thereby also increasing MRTF nuclear localization and increasing MRTF/SRF activity^[46].

hMSCs grown on microfibrillar substrates displayed fewer focal adhesions than on films or TCP. In contrast, fibroblasts formed similar numbers of focal adhesions per cell area on microfibrillar substrates as on films or TCP. On TCP and films, the number of zyxin positive

focal adhesions was the same between hMSCs and fibroblasts. Knockdown of either zyxin or paxillin did not affect FGFR1 expression. In contrast to paxillin, zyxin knockdown is known to diminish stress fibers^[47-49]. While actin-myosin inhibition by blebbistatin did decrease FGFR1 expression, zyxin knockdown did not. Although fewer than normally, zyxin knockdown cells still form focal adhesions^[35, 50]. Our data suggests that the actin-myosin tension between these focal adhesions is still sufficient to maintain a higher FGFR1 expression, as full inhibition of actin-myosin by blebbistatin reduced FGFR1.

The reason for the difference between fibroblasts and hMSCs was not investigated here. Different cell types exhibit different cell spreading and traction forces in response to different substrate stiffnesses^[51-53]. Indeed, the optimal stiffness for differentiation and proliferation defer per cell type^[54, 55]. We have previously shown that hMSCs experience the microfibrillar substrates used here as a soft substrate, demonstrated by fewer focal adhesions, less lamin A/C and less YAP nuclear translocation^[35]. The difference in focal adhesion formation between hMSCs and fibroblasts observed here potentially derives from a different response to matrix stiffness. Perhaps fibroblasts are able to form focal adhesions on softer substrates than hMSCs. Side by side comparison of hMSCs and fibroblasts on different stiffnesses has not yet been reported but could shed light on the differences observed here. It is also possible that besides a difference in focal adhesion formation, other phenotypical differences between hMSCs and fibroblasts play a role in FGFR1 regulation and their response to bFGF. We have shown that both hMSCs and fibroblasts regulate FGFR1 through MRTF/SRF and actin-myosin, but how MRTF/SRF and actin-myosin are regulated could differ significantly between the two cell-types. Such potential differences have not yet been thoroughly investigated. Additionally, other proteins that are differently expressed between hMSCs and fibroblasts could also affect FGFR1 expression. A deeper investigation in different proteins that regulate FGFR1 and how they are regulated would aid in explaining the observed differences between hMSCs and fibroblasts. For example, DNA pull down of FGFR1 promotor regions could be used to identify novel transcription factors. The role of mechanosensitive pathways in FGFR1 regulation that we have shown here could then be used to identify how these novel transcription factors are regulated.

The regulation of FGFR1 expression is, however, poorly studied. With the inhibitors of MRTF/SRF and actin-myosin, we could show that these protein-complexes greatly influence FGFR1 expression. It is of course possible that other factors also further influenced the

expression of FGFR1. This could be due to downstream effects of the change in actin-myosin, or through co-activation of independent pathways. YAP knockdown has been shown to decrease FGFR1 expression in lung cancer cells^[19] and neurospheres^[20]. Also, integrin $\alpha 6$ has been shown to regulate FGFR1^[20]. We found that YAP knockdown didn't alter FGFR1 expression in hMSCs, suggesting a different role for YAP in different cell types. YAP and integrin $\alpha 6$ regulation of FGFR1 does, however, hint at the mechanosensitive regulation of FGFR1, in accordance with what we've shown here. Other proven mechanisms of FGFR1 regulation include regulation by Pdx-1^[56, 57] and ZEB1^[20]. Whether these proteins play a role in FGFR1 regulation on microfibillar substrates has not been investigated here. The regulation of FGFR1 by MRTF/SRF and actin-myosin tension presented here adds to the sparse literature on FGFR1 regulation.

These novel findings can give insight in tumor development, as aberrant FGFR1 regulation is important in a wide variety of cancers^[3, 8]. FGFR inhibitors are already being used in clinical trials as novel anti-cancer drugs^[9-14]. Our study opens up new potential targets for FGFR1 regulation in cancer cells. Also, as an important regulator of proliferation in hMSCs^[18] and other cell types^[58, 59], this can have implications for scaffold designs. We show here that the scaffold design itself, as well as material properties, can influence the FGFR1 expression. Optimizing scaffold design to influence MRTF/SRF activity and FGFR1 expression could be crucial for certain tissue regeneration applications.

Conclusion

Microfibrillar substrates were successfully functionalized with bFGF, which increased the proliferation of fibroblasts, but not hMSCs. hMSCs, but not fibroblasts, reduced FGFR1 expression on microfibrillar substrates, explaining the lack of response to bFGF on microfibrillar substrates. Fibroblasts maintained a high expression of FGFR1 on microfibrillar substrates, explaining the difference in bFGF responsiveness between hMSCs and fibroblasts. hMSCs, but not fibroblasts, displayed fewer focal adhesions and expressed less SRF on microfibrillar substrates than on TCP or 2D film controls. In hMSCs and fibroblasts, the inhibition of actin-myosin interaction and MRTF/SRF activity decreased FGFR1 expression. In osteosarcoma MG63 cells, MRTF/SRF inhibition also led to decreased FGFR1 expression. Together, our data shows that hMSCs become irresponsive to bFGF on microfibrillar substrates because of a downregulation of SRF, which leads to a decrease in

FGFR1. Fibroblasts maintain a high SRF and FGFR1 expression and remain responsive to bFGF on microfibrillar substrates.

Methods

Film and microfibrillar substrate production

Random block co-polymer of poly(ethylene oxide terephthalate) (PEOT) and poly(butylene terephthalate) (PBT), with 300 Da PEO and PEOT/PBT ratio (w/w) of 55/45 (300PEOT55PBT45, acquired from PolyVation) was used to produce films and microfibrillar substrates. 300PEOT55/PBT45 granules were melted at 180 °C under slight pressure (~100 kg) in a circular 23 mm mold between two silicon wafers (Si-mat, Kaufering, Germany) to produce flat films. Films were punched out using a 22 mm punch to fit in a 12 well plate.

The electrospinning polymer solution was prepared by dissolving 20% (w/v) 300PEOT55PBT45 in 3:7 1,1,1,3,3,3-Hexafluoro-2-propanol (HFIP):chloroform, overnight at room temperature under agitation. For heparin functionalization, 2% (w/v) poly(ethylene glycol) (PEG) with 2 NH₂ end-groups (Mw: 3350 kDa) (PEG-NH₂), was added to the polymer solution and mixed for 4 hours before electrospinning.

ESP scaffolds (microfibrillar substrates) were produced on a slowly rotating (100 RPM) 19 cm diameter mandrel by electrospinning on a polyester mesh (FinishMat 6691 LL (40 gr/m²), generously provided by Lantor B.V.) with 12 mm holes, on top of aluminum foil. The following parameters were maintained: 15 cm working distance between needle and rotating mandrel, 1 ml/h flow rate, 23-25 °C and 40% relative humidity, a needle charge between 10-15 kV and collector charge between -2 and -5 kV. Individual ESP scaffolds were punched out with a diameter of 15 mm over the 12 mm holes in the polyester mesh and removed from the aluminum foil. This resulted in 15 mm ESP scaffolds with a 12 mm diameter surface for cell culture and a 1.5 mm polyester ring around it to improve handleability. Using this method, up to 100 microfibrillar substrates were produced under exactly equal parameters.

Before cell culture, microfibrillar substrates and films were sterilized in 70% ethanol for 15 min and dried at room temperature until visually dry. The 1.5 mm polyester ring was covered with a rubber 15 mm outer- and 12 mm inner-diameter O-ring (Eriks) to keep the scaffolds from floating in tissue culture well plates.

Functionalization of microfibrillar substrates with BSA or bFGF

Before coupling of bovine serum albumin (BSA)-FITC conjugate (ThermoFisher Scientific) or basic fibroblast growth factor (bFGF) (Neuromics), ethanol sterilized microfibrillar substrates

were incubated in 0.5 M NaOH for 30 min at room temperature to open the ester bond of the 300PEOT55PBT45 polymer. Scaffolds were thoroughly washed 5 times with water and then incubated with 4 mg/ml N-(3-Dimethylaminopropyl)-N'-ethylcarbodiimide hydrochloride (EDC) (Sigma-Aldrich) and 10 mg/ml N-hydroxysuccinimide (NHS) (Sigma-Aldrich) in milliQ water, or in milliQ water only, without EDC-NHS, as negative control, for 30 min at room temperature on a rocking plate. EDC-NHS solution was removed and 500 µl of 1 µg/ml BSA or 20, 200, or 2000 ng/ml bFGF in water was added to the scaffolds in a 24 well-plate well and incubated overnight at 4 °C on a rocking plate. The following day, BSA-FITC scaffolds were washed 5 times with water and scaffold fluorescence was measured in the fluorescein channel on a Clariostar plate reader (BMG Labtech). For sodium dodecyl sulfate (SDS) wash, 1% (w/v) in water was added to the functionalized scaffolds and incubated under agitation at room temperature overnight. The following day, scaffolds were thoroughly washed 5 times with water and measured on the plate reader as described before.

For bFGF functionalized scaffolds, bFGF solution was harvested to be analyzed by bFGF ELISA, and the scaffolds were washed 5 times with water, once with PBS and once with medium. For the bFGF scaffolds, all solutions were sterilized by filtration through at 0.2 µm filter. bFGF was quantified using a bFGF ELISA kit (Abcam), according to manufacturer's protocol.

Heparin functionalization of microfibrillar substrates

1.5 mg/ml heparin sodium salt from porcine intestinal mucosa (Sigma-Aldrich) was mixed with 4 mg/ml EDC and 10 mg/ml NHS in water (or water only, without EDC-NHS, as negative control) and directly added to the 300PEOT55PBT45+PEG-NH₂ microfibrillar substrates and incubated overnight at 4 °C.

To measure bound heparin, scaffolds were washed with 5 times with milliQ water and stained for 30 min with alcian blue staining solution (0.1% alcian blue, 10% ethanol, 0.1% acetic acid, 0.03 M MgCl₂ in water (all Sigma-Aldrich)). Scaffolds were washed once with MQ water and incubated for 30 min at room temperature in destaining solution (10% ethanol, 0.1% acetic acid, 0.03 M MgCl₂ in water). Scaffolds were washed again once with water and then incubated for 30 min in 1% SDS to extract the heparin-bound alcian blue from the scaffolds. The absorbance of this solution was measured in a Clariostar plate reader.

Comment [A21]: The figure says 10, 100, 1000 ng – am I missing something?

For cell culture, the heparin functionalized scaffolds were washed 5 times with milliQ water and incubated overnight at 4 °C with 500 µl 2000 ng/ml bFGF. The following day scaffolds were washed 5 times with water, once with PBS and once with medium. All solutions were sterilized by filtration through a 0.2 µm filter.

Cell culture

Human dermal fibroblasts (Lonza) were expanded at 2000 cells/cm² in DMEM+Glutamax medium (Thermo Fisher Scientific) supplemented with 10% (V/V) fetal bovine serum (FBS) (Sigma-Aldrich). Bone marrow derived hMSCs were isolated by Texas A&M Health Science Center^[60]. Briefly, aspirated bone marrow was centrifuged to isolate mononuclear cells. The hMSCs were further expanded and tested for differentiation potential. hMSCs were received at passage 1 and were further expanded at 1000 cells/cm² in α-MEM+Glutamax medium (Thermo Fisher Scientific) supplemented with 10% FBS. MG63 cells (ATCC) were expanded at 5000 cells/cm² in DMEM+Glutamax+10% FBS medium. All cells were cultured in 37 °C in 5% CO₂ until reaching 70-80% confluency. Cells were trypsinised in 0.05% Trypsin and 0.53 mM EDTA (ThermoFisher Scientific) and hMSCs and fibroblasts were used for experiments at passage 5. MG63 cells were used at passage 90.

Unless otherwise stated, all experiments were harvested at day 7. For scaffold experiments, hMSCs and fibroblasts were cultured at 1000 cells/cm² in TCP and films, and 30.000 cells/microfibrillar substrate in growth medium with and 100 U/ml penicillin-streptomycin. All other experiments were done in medium without penicillin-streptomycin. In bFGF medium conditions, 10 ng/ml bFGF was added to the medium.

For blebbistatin and MRTF/SRF inhibitor experiments, hMSCs and fibroblasts cells were seeded at 1000 cells/cm² on TCP and cultured for 6 days. MG63 cells were seeded at 5000 cells/cm² cultured for 2 days, because of a very high proliferation rate. After the initial culture period in growth medium, 100 µM blebbistatin (Sigma-Aldrich) in 0.2% DMSO in growth medium, or 12.2 µM MRTF/SRF inhibitor CCG203971 in 0.1% DMSO in growth medium, or respective DMSO control was added to the cells for 24 h.

To test the responsiveness of hMSCs to bFGF in the presence of MRTF/SRF inhibitor, hMSCs were seeded at 1000 cells/cm² in TCP and cultured for 7 days in 0.1% DMSO control, 0.1% DMSO + 10 ng/ml bFGF, 24.4 µM MRTF/SRF in 0.1% DMSO or 24.4 µM MRTF/SRF in 0.1% DMSO + 10 ng/ml bFGF, all in hMSC growth medium.

DNA quantification

To lyse cells for DNA quantification, cells were washed 2x with PBS and freeze-thawed dry twice before RLT lysis buffer (Qiagen) was added. Microfibrillar substrates were removed from the polyester ring after the last PBS wash. Samples were then freeze-thawed 3x in lysis buffer. TCP plates and films were scraped with a cell scraper after the first freeze-thaw in lysis buffer. Microfibrillar substrates were left in lysis buffer. Samples were diluted 100-400x, depending on expected number of cells per samples, in Tris-EDTA buffer (10 mM Tris-HCl, 1 mM EDTA, pH 7.5) and λ -DNA standard was made in the same final solution (0.25-1% RLT in Tris-EDTA buffer). Pico green assay (ThermoFisher Scientific) was then used to quantify DNA, according to manufacturer's protocol.

Protein isolation and western blot

Protein was isolated in a custom lysis buffer to allow for the detection of membrane proteins with western blot. Other buffers, such as RIPA buffer, were tried for FGFR1 western blot without success (data not shown). The buffer consisted of 150 mM NaCl, 0.5% sodium deoxycholate, 1% SDS, 1% NP-40 and 50 mM Tris-HCl in water, set to pH 7.4. The buffer was supplemented with cComplete™ Mini EDTA-free Protease Inhibitor Cocktail (Sigma-Aldrich). Samples were washed in cold PBS twice before lysis. Microfibrillar substrates were removed from the supporting polyester ring. To get sufficient proteins, 6–12 films or 15–20 microfibrillar substrates were combined in 300-400 μ l for a single protein isolate. Experiments were repeated 3 or 4 times to obtain sufficient replicates. 6 or 10 cm dishes were used for TCP samples. TCP and film conditions were scraped in lysis buffer with cell scrapers. Microfibrillar substrates were submerged in lysis buffer and incubated for around 30 min in lysis buffer because the scaffolds were removed from the protein isolate. Samples were not spun down to maintain potentially non-dissolved membrane proteins in solution. Pierce BCA protein assay kit (Thermo Fisher Scientific) was used to quantify total protein concentration. 20 μ g protein was incubated in 10% 2-Mercaptoethanol (Sigma-Aldrich) in Laemmli loading buffer (Bio-Rad) for 37 °C for 20 min for FGFR1 western blots and at 95 °C for 5 min for all other western blots. Samples were loaded into 4–15% polyacrylamide gels (Bio-Rad) and blotted to 0.45 μ m PVDF membranes (Bio-Rad) using semi-dry transfer. Membranes were blocked in 5% (w/v) fat free milk (Bio-Rad) in TBS + 0.05% (v/v) tween-20

(Sigma-Aldrich) for 1 hour, except for SRF western blots, which had to be blocked in 2% (w/v) BSA (VWR) + 0.05% tween-20 in PBS to work (data not shown). Primary antibodies were incubated in their respective blocking buffer overnight at 4 °C. All antibodies were ordered from Abcam: FGFR1: ab76464 1/500; Paxillin: ab32084 1/1000; Zyxin: ab58210 1/1000; YAP1: ab52771 1/1000; SRF: ab53147 1/250; TBP: ab51841 1/1000. Blots were incubated the following day with 0.33 µg/ml Goat-anti-rabbit or -mouse HRP (Bio-Rad) in blocking buffer for 1 h at room temperature. To visualize the protein bands, blots were incubated with Clarity Western ECL (Bio-Rad) for 1-5 min right before imaging.

Immunofluorescence and imaging

Cells were fixed with 3.6% (v/v) paraformaldehyde (Sigma-Aldrich) in PBS for 20 min at room temperature. To block and permeabilize, fixed cells were incubated in 2% (w/v) BSA+0.1% (v/v) triton X (VWR) in PBS. Zyxin or paxillin (Abcam, ab58210 and ab32084, respectively, both 1/1000) were incubated in 2% (w/v) BSA+0.05% (v/v) tween-20 in PBS overnight at 4 °C. The following day, 1/1000 Goat-anti-mouse Alexa Fluor 568 or Goat-anti-rabbit Alexa Fluor 488 was incubated overnight at 4 °C in 2% (w/v) BSA+0.05% (v/v) tween-20 in PBS. The next day, samples were stained with DAPI (Sigma-Aldrich, 0.14 µg/ml in PBS+0.05% (v/v) tween-20) to stain nuclei. Images were taken on a confocal microscope.

Focal adhesions were quantified manually by counting the number of focal adhesions per cell using Fiji. Between 17 and 27 cells were counted per condition, from 5-10 different images from biological triplicates. Cell area was measured by manually outlining the cells and measuring surface area using Fiji. The number of focal adhesions was normalized to the cell area.

Lentiviral production and transduction

To produce lentiviral particles, human embryonic kidney 293FT (HEK) cells were seeded at 60.000 cells/cm² in DMEM+Glutamax+10% FBS. Cells were transfected with pMDLg pRRE, pMD2.G, pRSV Rev (Addgene) and one of the pLKO.1 shRNA plasmids using 5:1 lipofectamine 2000 (Thermo Fisher Scientific):DNA (v/w) 24 h after seeding. The following TRC pLKO.1 constructs (Dharmacon) were used: ZYX: TRCN0000074204 and TRCN0000074205; PXN: TRCN0000123134 and TRCN0000123136; YAP1: TRCN0000107265

and TRCN0000107266; and non-targeting shRNA control (RHS6848). Medium was changed 16 h post-transfection to hMSC growth medium. Lentivirus was harvested and filtered through a 0.45 μm filter 24 h and 48 h after the change to hMSC growth medium.

24 h after thawing at 1000 cells/cm², hMSCs were transduced with the lentiviral medium for 16 h. Medium was replaced with growth medium the following day. 48-72 h post transduction, medium was replaced with growth medium + 2 $\mu\text{g}/\text{ml}$ puromycin for 72 hours. A total of 9-10 days after thawing, hMSCs were passaged and seeded at 1000 cells/cm² on TCP for 7 days in growth medium before protein harvest.

Statistics

The statistical tests and number of biological replicates and/or experiments are stated in the figure subtexts. Each experiment used at least 3 biological replicas. Cells selected for quantification of focal adhesions were selected randomly. Films and electrospun scaffolds were also randomly assigned to different experimental groups. Shapiro-Wilk test was used to test for normal distribution of each experimental group before further statistical analysis. To test for significance of absolute differences in experiments with multiple comparisons between groups, a One-way ANOVA with Tukey's post hoc was performed. For relative differences between multiple experimental groups, log values were used for repeated measures ANOVA, with Tukey's post-hoc test. For experiments with a single comparison, two-tailed student's t-test was used for absolute differences, and ratio-paired t-test for relative differences. Significance was set at $p < 0.05$. Statistical analysis was done using Graphpad Prism 8.

Acknowledgements

We would like to thank Matt Baker and Paul Wieringa for the valuable discussions about the functionalization strategy of the electrospun scaffolds. We are grateful to the European Research Council starting grant "Cell Hybridize" for financial support under the Horizon2020 framework program (Grant #637308). Some of the materials used in this work were provided by the Texas A&M Health Science Center College of Medicine Institute for Regenerative Medicine at Scott & White through a grant from NCRR of the NIH (Grant #P40RR017447).

References

1. A. Bikfalvi, S. Klein, G. Pintucci, and D.B. Rifkin, *Biological roles of fibroblast growth factor-2*. *Endocr Rev*, 1997. **18**(1): p. 26-45.
2. D.M. Ornitz and N. Itoh, *The Fibroblast Growth Factor signaling pathway*. Wiley Interdiscip Rev Dev Biol, 2015. **4**(3): p. 215-66.
3. T. Helsten, S. Elkin, E. Arthur, B.N. Tomson, J. Carter, and R. Kurzrock, *The FGFR Landscape in Cancer: Analysis of 4,853 Tumors by Next-Generation Sequencing*. *Clin Cancer Res*, 2016. **22**(1): p. 259-67.
4. H.S. Kim, J.H. Kim, H.J. Jang, B. Han, and D.Y. Zang, *Pathological and Prognostic Impacts of FGFR2 Overexpression in Gastric Cancer: A Meta-Analysis*. *J Cancer*, 2019. **10**(1): p. 20-27.
5. M.N. Fletcher, M.A. Castro, X. Wang, I. de Santiago, M. O'Reilly, S.F. Chin, O.M. Rueda, C. Caldas, B.A. Ponder, F. Markowitz, and K.B. Meyer, *Master regulators of FGFR2 signalling and breast cancer risk*. *Nat Commun*, 2013. **4**: p. 2464.
6. S. Tang, Y. Hao, Y. Yuan, R. Liu, and Q. Chen, *Role of fibroblast growth factor receptor 4 in cancer*. *Cancer Sci*, 2018. **109**(10): p. 3024-3031.
7. R. Fuller, *Cardiac function and the neonatal EKG. Part I: Introduction to neonatal EKGs*. *Neonatal Netw*, 1989. **7**(4): p. 47-51.
8. J. Wesche, K. Haglund, and E.M. Haugsten, *Fibroblast growth factors and their receptors in cancer*. *Biochem J*, 2011. **437**(2): p. 199-213.
9. Y.K. Chae, K. Ranganath, P.S. Hammerman, C. Vaklavas, N. Mohindra, A. Kalyan, M. Matsangou, R. Costa, B. Carneiro, V.M. Villaflor, M. Cristofanilli, and F.J. Giles, *Inhibition of the fibroblast growth factor receptor (FGFR) pathway: the current landscape and barriers to clinical application*. *Oncotarget*, 2017. **8**(9): p. 16052-16074.
10. D. Piasecka, M. Braun, K. Kitowska, K. Mieczkowski, R. Kordek, R. Sadej, and H. Romanska, *FGFs/FGFRs-dependent signalling in regulation of steroid hormone receptors - implications for therapy of luminal breast cancer*. *J Exp Clin Cancer Res*, 2019. **38**(1): p. 230.
11. N. Sobhani, A. Ianza, A. D'Angelo, G. Roviello, F. Giudici, M. Bortul, F. Zanconati, C. Bottin, and D. Generali, *Current Status of Fibroblast Growth Factor Receptor-Targeted Therapies in Breast Cancer*. *Cells*, 2018. **7**(7).
12. S.K. Pal, J.E. Rosenberg, J.H. Hoffman-Censits, R. Berger, D.I. Quinn, M.D. Galsky, J. Wolf, C. Dittrich, B. Keam, J.P. Delord, J.H.M. Schellens, G. Gravis, J. Medioni, P. Maroto, V. Sriuranpong, C. Charoentum, H.A. Burris, V. Grunwald, D. Petrylak, U. Vaishampayan, E. Gez, U. De Giorgi, J.L. Lee, J. Voortman, S. Gupta, S. Sharma, A. Mortazavi, D.J. Vaughn, R. Isaacs, K. Parker, X. Chen, K. Yu, D. Porter, D. Graus Porta, and D.F. Bajorin, *Efficacy of BGI398, a Fibroblast Growth Factor Receptor 1-3 Inhibitor, in Patients with Previously Treated Advanced Urothelial Carcinoma with FGFR3 Alterations*. *Cancer Discov*, 2018. **8**(7): p. 812-821.
13. C. Heinzle, Z. Erdem, J. Paur, B. Grasl-Kraupp, K. Holzmann, M. Grusch, W. Berger, and B. Marian, *Is fibroblast growth factor receptor 4 a suitable target of cancer therapy?* *Curr Pharm Des*, 2014. **20**(17): p. 2881-98.
14. L. Lang and Y. Teng, *Fibroblast Growth Factor Receptor 4 Targeting in Cancer: New Insights into Mechanisms and Therapeutic Strategies*. *Cells*, 2019. **8**(1).
15. T.E. Kahkonen, K.K. Ivaska, M. Jiang, K.G. Buki, H.K. Vaananen, and P.L. Harkonen, *Role of fibroblast growth factor receptors (FGFR) and FGFR like-1 (FGFRL1) in*

- mesenchymal stromal cell differentiation to osteoblasts and adipocytes*. Mol Cell Endocrinol, 2018. **461**: p. 194-204.
16. H. Miraoui, K. Oudina, H. Petite, Y. Tanimoto, K. Moriyama, and P.J. Marie, *Fibroblast growth factor receptor 2 promotes osteogenic differentiation in mesenchymal cells via ERK1/2 and protein kinase C signaling*. J Biol Chem, 2009. **284**(8): p. 4897-904.
 17. X. Wen, X. Li, Y. Tang, J. Tang, S. Zhou, Y. Xie, J. Guo, J. Yang, X. Du, N. Su, and L. Chen, *Chondrocyte FGFR3 Regulates Bone Mass by Inhibiting Osteogenesis*. J Biol Chem, 2016. **291**(48): p. 24912-24921.
 18. C. Dombrowski, T. Helledie, L. Ling, M. Grunert, C.A. Canning, C.M. Jones, J.H. Hui, V. Nurcombe, A.J. van Wijnen, and S.M. Cool, *FGFR1 signaling stimulates proliferation of human mesenchymal stem cells by inhibiting the cyclin-dependent kinase inhibitors p21(Waf1) and p27(Kip1)*. Stem Cells, 2013. **31**(12): p. 2724-36.
 19. T. Lu, Z. Li, Y. Yang, W. Ji, Y. Yu, X. Niu, Q. Zeng, W. Xia, and S. Lu, *The Hippo/YAP1 pathway interacts with FGFR1 signaling to maintain stemness in lung cancer*. Cancer Lett, 2018. **423**: p. 36-46.
 20. A. Kowalski-Chauvel, V. Gouaze-Andersson, L. Baricault, E. Martin, C. Delmas, C. Toulas, E. Cohen-Jonathan-Moyal, and C. Seva, *Alpha6-Integrin Regulates FGFR1 Expression through the ZEB1/YAP1 Transcription Complex in Glioblastoma Stem Cells Resulting in Enhanced Proliferation and Stemness*. Cancers (Basel), 2019. **11**(3).
 21. S. Dupont, L. Morsut, M. Aragona, E. Enzo, S. Giulitti, M. Cordenonsi, F. Zanconato, J. Le Digabel, M. Forcato, S. Bicciato, N. Elvassore, and S. Piccolo, *Role of YAP/TAZ in mechanotransduction*. Nature, 2011. **474**(7350): p. 179-83.
 22. Z. Sun, S.S. Guo, and R. Fassler, *Integrin-mediated mechanotransduction*. J Cell Biol, 2016. **215**(4): p. 445-456.
 23. M. Barczyk, S. Carracedo, and D. Gullberg, *Integrins*. Cell Tissue Res, 2010. **339**(1): p. 269-80.
 24. M.A. Wozniak, K. Modzelewska, L. Kwong, and P.J. Keely, *Focal adhesion regulation of cell behavior*. Biochim Biophys Acta, 2004. **1692**(2-3): p. 103-19.
 25. M.J. Stroud, *Linker of nucleoskeleton and cytoskeleton complex proteins in cardiomyopathy*. Biophys Rev, 2018. **10**(4): p. 1033-1051.
 26. K. Weber and U. Groeschel-Stewart, *Antibody to myosin: the specific visualization of myosin-containing filaments in nonmuscle cells*. Proc Natl Acad Sci U S A, 1974. **71**(11): p. 4561-4.
 27. C.S. Chen, M. Mrksich, S. Huang, G.M. Whitesides, and D.E. Ingber, *Geometric control of cell life and death*. Science, 1997. **276**(5317): p. 1425-8.
 28. D.R. Croft and M.F. Olson, *The Rho GTPase effector ROCK regulates cyclin A, cyclin D1, and p27Kip1 levels by distinct mechanisms*. Mol Cell Biol, 2006. **26**(12): p. 4612-27.
 29. N. Takeda, M. Kondo, S. Ito, Y. Ito, K. Shimokata, and H. Kume, *Role of RhoA inactivation in reduced cell proliferation of human airway smooth muscle by simvastatin*. Am J Respir Cell Mol Biol, 2006. **35**(6): p. 722-9.
 30. L.C. Boraas, E.T. Pineda, and T. Ahsan, *Actin and myosin II modulate differentiation of pluripotent stem cells*. PLoS One, 2018. **13**(4): p. e0195588.
 31. C.P. Heisenberg and Y. Bellaïche, *Forces in tissue morphogenesis and patterning*. Cell, 2013. **153**(5): p. 948-62.
 32. A.J. Steward and D.J. Kelly, *Mechanical regulation of mesenchymal stem cell differentiation*. J Anat, 2015. **227**(6): p. 717-31.

33. A.C. Callan-Jones and R. Voituriez, *Actin flows in cell migration: from locomotion and polarity to trajectories*. *Curr Opin Cell Biol*, 2016. **38**: p. 12-7.
34. E.N. Olson and A. Nordheim, *Linking actin dynamics and gene transcription to drive cellular motile functions*. *Nat Rev Mol Cell Biol*, 2010. **11**(5): p. 353-65.
35. J. Zonderland, I.L. Moldero, S. Anand, C. Mota, and L. Moroni, *Dimensionality changes actin network through lamin A/C and zyxin*. *Biomaterials*, 2020. **240**: p. 119854.
36. S. Lotz, S. Goderie, N. Tokas, S.E. Hirsch, F. Ahmad, B. Corneo, S. Le, A. Banerjee, R.S. Kane, J.H. Stern, S. Temple, and C.A. Fasano, *Sustained levels of FGF2 maintain undifferentiated stem cell cultures with biweekly feeding*. *PLoS One*, 2013. **8**(2): p. e56289.
37. T.H. Nguyen, S.H. Kim, C.G. Decker, D.Y. Wong, J.A. Loo, and H.D. Maynard, *A heparin-mimicking polymer conjugate stabilizes basic fibroblast growth factor*. *Nat Chem*, 2013. **5**(3): p. 221-7.
38. S.J. Paluck, T.H. Nguyen, J.P. Lee, and H.D. Maynard, *A Heparin-Mimicking Block Copolymer Both Stabilizes and Increases the Activity of Fibroblast Growth Factor 2 (FGF2)*. *Biomacromolecules*, 2016. **17**(10): p. 3386-3395.
39. E.K.A. Nur, I. Ahmed, J. Kamal, A.N. Babu, M. Schindler, and S. Meiners, *Covalently attached FGF-2 to three-dimensional polyamide nanofibrillar surfaces demonstrates enhanced biological stability and activity*. *Mol Cell Biochem*, 2008. **309**(1-2): p. 157-66.
40. M.A. Nugent and R.V. Iozzo, *Fibroblast growth factor-2*. *Int J Biochem Cell Biol*, 2000. **32**(2): p. 115-20.
41. K.M. Hutchings, E.M. Lisabeth, W. Rajeswaran, M.W. Wilson, R.J. Sorenson, P.L. Campbell, J.H. Ruth, A. Amin, P.S. Tsou, J.R. Leipprandt, S.R. Olson, B. Wen, T. Zhao, D. Sun, D. Khanna, D.A. Fox, R.R. Neubig, and S.D. Larsen, *Pharmacokinetic optimization of CCG-203971: Novel inhibitors of the Rho/MRTF/SRF transcriptional pathway as potential antifibrotic therapeutics for systemic scleroderma*. *Bioorg Med Chem Lett*, 2017. **27**(8): p. 1744-1749.
42. E.M. Lisabeth, D. Kahl, I. Gopallawa, S.E. Haynes, S.A. Misesek, P.L. Campbell, T.S. Dexheimer, D. Khanna, D.A. Fox, X. Jin, B.R. Martin, S.D. Larsen, and R.R. Neubig, *Identification of Pirin as a Molecular Target of the CCG-1423/CCG-203971 Series of Antifibrotic and Antimetastatic Compounds*. *ACS Pharmacology & Translational Science*, 2019. **2** (2): p. 92-100.
43. H. Lee, S. Lim, M.S. Birajdar, S.H. Lee, and H. Park, *Fabrication of FGF-2 immobilized electrospun gelatin nanofibers for tissue engineering*. *Int J Biol Macromol*, 2016. **93**(Pt B): p. 1559-1566.
44. J.A. Spencer and R.P. Misra, *Expression of the SRF gene occurs through a Ras/Sp/SRF-mediated-mechanism in response to serum growth signals*. *Oncogene*, 1999. **18**(51): p. 7319-27.
45. C. Esnault, A. Stewart, F. Gualdrini, P. East, S. Horswell, N. Matthews, and R. Treisman, *Rho-actin signaling to the MRTF coactivators dominates the immediate transcriptional response to serum in fibroblasts*. *Genes & development*, 2014. **28**(9): p. 943-958.
46. D. Gau and P. Roy, *SRF'ing and SAP'ing - the role of MRTF proteins in cell migration*. *J Cell Sci*, 2018. **131**(19).

47. E. Griffith, A.S. Coutts, and D.M. Black, *RNAi knockdown of the focal adhesion protein TES reveals its role in actin stress fibre organisation*. *Cell Motil Cytoskeleton*, 2005. **60**(3): p. 140-52.
48. M.A. Smith, E. Blankman, M.L. Gardel, L. Luetjohann, C.M. Waterman, and M.C. Beckerle, *A zyxin-mediated mechanism for actin stress fiber maintenance and repair*. *Dev Cell*, 2010. **19**(3): p. 365-76.
49. A.A. Birukova, I. Cokic, N. Moldobaeva, and K.G. Birukov, *Paxillin is involved in the differential regulation of endothelial barrier by HGF and VEGF*. *Am J Respir Cell Mol Biol*, 2009. **40**(1): p. 99-107.
50. Z. Sun, S. Huang, Z. Li, and G.A. Meininger, *Zyxin is involved in regulation of mechanotransduction in arteriole smooth muscle cells*. *Front Physiol*, 2012. **3**: p. 472.
51. H.B. Wang, M. Dembo, and Y.L. Wang, *Substrate flexibility regulates growth and apoptosis of normal but not transformed cells*. *Am J Physiol Cell Physiol*, 2000. **279**(5): p. C1345-50.
52. P.C. Georges and P.A. Janmey, *Cell type-specific response to growth on soft materials*. *J Appl Physiol* (1985), 2005. **98**(4): p. 1547-53.
53. S.Y. Tee, A.R. Bausch, and P.A. Janmey, *The mechanical cell*. *Curr Biol*, 2009. **19**(17): p. R745-8.
54. H. Lv, L. Li, M. Sun, Y. Zhang, L. Chen, Y. Rong, and Y. Li, *Mechanism of regulation of stem cell differentiation by matrix stiffness*. *Stem Cell Res Ther*, 2015. **6**: p. 103.
55. R.G. Wells, *The role of matrix stiffness in regulating cell behavior*. *Hepatology*, 2008. **47**(4): p. 1394-400.
56. H. Wang, M. Iezzi, S. Theander, P.A. Antinozzi, B.R. Gauthier, P.A. Halban, and C.B. Wollheim, *Suppression of Pdx-1 perturbs proinsulin processing, insulin secretion and GLP-1 signalling in INS-1 cells*. *Diabetologia*, 2005. **48**(4): p. 720-31.
57. A.W. Hart, N. Baeza, A. Apelqvist, and H. Edlund, *Attenuation of FGF signalling in mouse beta-cells leads to diabetes*. *Nature*, 2000. **408**(6814): p. 864-8.
58. K. Wang, W. Ji, Y. Yu, Z. Li, X. Niu, W. Xia, and S. Lu, *FGFR1-ERK1/2-SOX2 axis promotes cell proliferation, epithelial-mesenchymal transition, and metastasis in FGFR1-amplified lung cancer*. *Oncogene*, 2018. **37**(39): p. 5340-5354.
59. C. Fumarola, N. Bozza, R. Castelli, F. Ferlenghi, G. Marseglia, A. Lodola, M. Bonelli, S. La Monica, D. Cretella, R. Alfieri, R. Minari, M. Galetti, M. Tiseo, A. Ardizzoni, M. Mor, and P.G. Petronini, *Expanding the Arsenal of FGFR Inhibitors: A Novel Chloroacetamide Derivative as a New Irreversible Agent With Anti-proliferative Activity Against FGFR1-Amplified Lung Cancer Cell Lines*. *Front Oncol*, 2019. **9**: p. 179.
60. C.M. Digirolamo, D. Stokes, D. Colter, D.G. Phinney, R. Class, and D.J. Prockop, *Propagation and senescence of human marrow stromal cells in culture: a simple colony-forming assay identifies samples with the greatest potential to propagate and differentiate*. *Br J Haematol*, 1999. **107**(2): p. 275-81.

Reviewers' comments:

Reviewer #1 (Remarks to the Author):

Given the current Covid-19 situation it is totally understandable that no additional experiments were possible, but the authors have done a great job in restructuring the article and improving the clarity of the manuscript. All points that were raised by the reviewers were addressed satisfactorily and I could not find any parts that would need correction or clarification.

I would recommend the article for publication.

Reply: We thank the reviewer for their kind words.

Reviewer #2 (Remarks to the Author):

The revised version of this manuscript from the Moroni's group addressed most of my criticism. I am particularly impressed with the molecular biology in this study but fabrication and characterization details, as well as the overall clarity and writing of the manuscript could be improved.

Reply: We thank the reviewer for the kind words and the detailed look to our manuscript. We've replied to the comments below and we hope we have clarified a few points sufficiently. We have further adapted the manuscript and hope we have sufficiently addressed the critiques of the reviewer.

I am attaching a revised word document (the name of the reviewer was removed) to indicate a number of possible useful changes.

My two main criticisms are as follows:

1. The connection with cancer is really confusing. Entire paragraphs in the Intro and Discussion are devoted to cancer but, at present, there are only two panels in a supplemental figure worth of results obtained from a single cell line. If the connection with cancer is important, more data is needed. If not, the storyline can be removed entirely and the introduction and discussion can be dedicated only to the regenerative medicine storyline. Importantly, regenerative medicine is a stronger application of this research since fibrous scaffolds are often employed (alone or in combination with cells) as regenerative interventions.

Reply: We agree with the reviewer that the main story line is and should be about regenerative medicine. We have therefore removed all cancer biology-related paragraphs from the abstract and the introduction. We still think the experiment with the MG63 cells is a nice addition to the manuscript, so we have left it in. Not necessarily for the link to cancer biology, but also because it is another cell type where we have shown an effect of the MRTF/SRF inhibitor on FGFR1 expression. We do think the discussion is the right place to elaborate on results and make connections with other study areas. As FGFR1 is so important for cancer biology, and its regulation greatly unstudied, we think it is nice to make a small leap in that direction, and at least mention it in the discussion. So, we have left and slightly adapted the paragraph in the discussion about potential uses of this study in the cancer field.

2. I am still confused by the quantification of bFGF tethering via ELISA (text and figure 1). As I indicated in the word document, it might be the writing. If I understand correctly, three known concentrations of bFGF were tested with and without EDC-NHS chemistry. The washout solution is then tested with ELISA to measure the bFGF that bound a-specifically and, therefore, was released. A few considerations:

Reply: The reviewer has misunderstood the method, which means that we didn't explain it clearly enough in the text. We have updated the text in the results and methods section and we hope that it is now clear to the reader. We'll also explain it here directly and a bit more elaborately to the reviewer: With the BSA-FITC we could directly measure the fluorescence to quantify the (relative) amount of BSA-FITC bound to the surface. With bFGF, we were not able to directly measure bFGF on the surface of the microfibrillar scaffolds (despite attempts with several different methods that are not in the manuscript). So, we chose to not measure the bFGF bound to the scaffolds, but the bFGF that did not bind to the scaffolds. To measure this, we did a bFGF ELISA on the bFGF solution with which the scaffolds were incubated. So: scaffolds were incubated with EDC/NHS solution (or water for negative control) and then all incubated with a bFGF solution with different concentrations. After overnight incubation, we took a bit from this bFGF solution that was on the scaffolds, so before any washes, and measured the concentration of bFGF in this solution. This allowed us to see how much bFGF was left over in solution,

and so how much did not bind to the scaffold. In this way, we saw that around 70% of bFGF bound to the scaffolds (30% of the original concentration was left over in solution) in each concentration in the scaffolds not activated with EDC/NHS. So if we put 500 μ l of 20 ng/ml (10 ng total) on top of the scaffolds, around 3 ng total was left over in the bFGF solution after incubation on the scaffolds. This can be seen in figure 1b. As this bFGF did not bind covalently (no EDC/NHS), we called this aspecific binding, meaning it just absorbed on the surface.

Then if we did activate the scaffolds with EDC/NHS before bFGF incubation, we saw that very little bFGF was left over in the bFGF incubation solution, around 2% was left over in that solution, meaning that around 98% has bound to the scaffolds. We called this quantity covalently binding, as the EDC/NHS clearly increased the amount of bound bFGF, meaning that at least part of that 98% covalently bound to the scaffold. How much that is, we don't know. To find that out one would have to directly measure the amount of bFGF on the surface of the microfibrillar scaffolds.

So, we did not measure the concentration of bFGF in the wash solutions, or afterwards. We did not do this because we extensively washed with a large amount of water/PBS and so concentrations of bFGF in the wash solutions were too low to measure by bFGF ELISA. Indeed, a follow up on this study could be a more thorough investigation on how much bFGF is covalently bound, how much is aspecifically bound, and how much aspecifically bound bFGF is released over how much time, etc. This would be a study on its own, since the main message of our manuscript is on FGFR1 regulation, and the bFGF coupling was a means to get to that answer. We believe the characterization here is sufficient to deliver that message, which is corroborated with data.

a. is there any bFGF left in the bathing solution before the washing steps? Was this tested with ELISA? If the largest known concentration in the bathing solution is 1000 ng but only 300 ng binds to the fibers, the left-over from ELISA should be normalized against this real value. I don't think this is the case (see point b) but it's unclear from the text.

Reply: See reply to point 2.

b. that the largest of the concentrations used (1000 ng) has essentially the same amount of un-bound bFGF after washout (\sim 0.1 ng from the graphs) then the smallest concentration (10 ng). Given the long incubation time (overnight) after EDC-NHS, I would expect most bFGF to bind which suggests 1000 ng is not sufficient to saturate the fibers. Or, alternatively (see point a) that not all 1000 ng bind before washout.

Reply: Indeed, as almost no bFGF is left in solution after EDC/NHS and 1000 ng bFGF, it could be that saturation is not reached yet. This point is added to the discussion.

c. since bFGF in solution was used at a concentration of 10 ng/mL (but we don't know the total volume) it's unclear how the tethered concentrations compare to the fiber deposited ones.

Reply: The concentration was 20 ng/ml with a total volume is 500 μ l, totaling to 10 ng per scaffold (for that condition). This is stated in the methods section.

The big problem here is that if bFGF is not saturating the fibers, an unknown percentage of the cells in that experimental group might not be exposed to bFGF which will invalidate the comparison. For example, if only 30% of the fiber surface area is coated but the DNA is collected from confluent cells, \sim 70% of the collected DNA will come from cells that were not exposed to bFGF.

Reply: Indeed, the percentage of the surface that is covered by bFGF is not known. However, the situation is a bit more complex than explained in the reviewer's comment. First of all, cells are many orders of magnitude larger than bFGF. If 30% (nm^2 cover by bFGF/ nm^2 total) of the surface is covered by bFGF, that would mean that for every square micrometer, 30% of that would be covered by bFGF, meaning that each cell would be exposed to many bFGF proteins. If the spacing between each bFGF would be (significantly) larger than a single cell, then indeed it's possible that some cells would not touch bFGF in a stationary situation. This also raises the question how many bFGF proteins a cell needs to be exposed to at any given time and/or over a period of time in order to increase proliferation. As far as we know, this has never been investigated. However, cells are not stationary and are constantly migrating or moving in the scaffold environment. Therefore, it would be highly unlikely that a cell would never encounter a bFGF protein. Of course, to increase proliferation the concentration of bFGF needs to be high enough so that the cell encounters enough bFGF proteins. In a situation where some cells do encounter enough bFGF to increase proliferation, and an X% of cells don't, we don't agree that would invalidate any comparison, it would only dilute the results. What we know for sure is that the fibroblasts population as a whole increased proliferation on the 1000 ng bFGF scaffolds. Whether this increase comes from a subset of fibroblasts or from every single fibroblast is not known. As they are primary cells at a low passage, they are inherently heterogeneous, making answering this question very difficult. A

certain subset will always have a higher proliferation rate, others might be destined for senescence, others more sensitive to bFGF or other growth factors in the medium, etc.

To answer the question: is the bFGF coupled to the microfibrillar scaffold functional? A 'population as a whole' increase in proliferation is enough to answer this question. We agree that it would be interesting to fully investigate the quantity, concentration, spacing, etc. of the bFGF on the scaffolds at each concentration. However, this would be a study on its own and not the focus of this study, as also explained in point 2.

We have added a section in the discussion to elaborate on this point and make suggestions for future studies.

Ultimately, I believe the experiments were done correctly but it is not clear from the current version of the manuscript. The authors should clarify the methodologies and the various steps used in this set of experiments. If additional ELISA experiments are still not possible, I would suggest a simple theoretical model of bFGF binding that can be used to estimate how much bFGF is needed to saturate the surface of the fibers. With such model, the experimental concentrations of bFGF used in the manuscript can be placed in context.

Reply: Calculating the saturation is unfortunately not as straightforward as suggested. One would need to know the surface area of the microfibrillar scaffolds. This is known to be very hard for electrospun scaffolds and we don't have this information for these scaffolds.

Before being able to understand the effect of a certain saturation, we would need to understand the amount of (bound) bFGF that is required to induce a certain cell response. We refer to our reply to the point above; we don't agree that this would be very relevant for this study, although we do agree that it would be interesting for studies that will fully and very thoroughly characterize functionalized scaffolds.

A number of minor comments and edits are in the attached revised document.

Reply: We have modified the text, accepted most of the suggested changes, and replied to the comments that were not yet answered in this document.

REVIEWERS' COMMENTS:

Reviewer #2 (Remarks to the Author):

Thank you for the extra work of clarifying the methods and the storyline! This is a very interesting piece of research and I look forward to seeing it published. Francesco Pasqualini